# Learning Shrinks the Hard Tail: Training-Dependent Inference Scaling in a Solvable Linear Model

**Noam Levi**
École Polytechnique Fédérale de Lausanne (EPFL)
Lausanne, Switzerland
`noam.levi@epfl.ch`

## Abstract

We analyze neural scaling laws in a solvable model of last-layer fine-tuning where targets have intrinsic, instance-heterogeneous difficulty. In our Latent Instance Difficulty (LID) model, each input's target variance is governed by a latent "precision" drawn from a heavy-tailed distribution. While generalization loss recovers standard scaling laws, our main contribution connects this to inference. The pass@$k$ failure rate exhibits a power-law decay, $k^{-\beta_{\text{eff}}}$, but the observed exponent $\beta_{\text{eff}}$ is training-dependent. It grows with sample size $N$ before saturating at an intrinsic limit $\beta$ set by the difficulty distribution's tail. This coupling reveals that learning shrinks the "hard tail" of the error distribution: improvements in the model's generalization error steepen the pass@$k$ curve until irreducible target variance dominates. The LID model yields testable, closed-form predictions for this behavior, including a compute-allocation rule that favors training before saturation and inference attempts after. We validate these predictions in simulations and in two real-data proxies: CIFAR-10H (human-label variance) and a maths teacher–student distillation task.

## 1 Introduction

The remarkable success of large-scale machine learning models is tightly linked to empirically observed and theoretically understood scaling laws, which characterize performance improvements with increasing data, model size, or training time (Kaplan et al., 2020; Hestness et al., 2017; Rosenfeld et al., 2019; Bahri et al., 2024; Maloney et al., 2022; Bordelon et al., 2024). These laws have been crucial for predicting learning curves and optimizing resource allocation in the large dataset size $N$ and number of parameters $P$ regime. Most classical analyses focus on the *training* side, relating generalization loss $\mathcal{L}_{\text{gen}}$ to $N$ (and sometimes $P$) through properties of the data distribution such as spectral decay (Bartlett et al., 2020; Hastie et al., 2020; Bordelon et al., 2020) or tractable model classes (Maloney et al., 2022; Tay et al., 2022).

A complementary paradigm emphasizes *inference-time* compute: repeated attempts, best-of-$N$, and search can produce substantial gains on difficult reasoning tasks even without further training, typically evaluated with pass@$k$ under a verifier (Snell et al., 2024; Brown et al., 2024). While practical methods for exploiting inference-time compute are rapidly evolving (see Sec. 2), and some explanations for their success have been given (Levi, 2024; Schaeffer et al., 2025a), a basic theoretical question remains: *how does training progress shape performance scaling at inference time?*

In many real-world settings the input–output relationship is not deterministic; some instances are intrinsically more variable than others (Arpit et al., 2017; Northcutt et al., 2021). Such instance-level heterogeneity affects both stages: during training we observe only a single noisy realization per input, and during inference we may compare a model prediction to multiple fresh realizations under pass@$k$. This motivates a minimal model that explicitly ties training and inference through *instance difficulty*.

**Setting.** We adopt a deliberately simple but analyzable framework that mirrors common practice in fine-tuning: *last-layer (linear) regression on fixed features* in high dimension, with intrinsically stochastic targets. Each instance $\mathbf{x}$ carries a latent *precision* $\tau_{\mathbf{x}}$ (its "difficulty"), drawn from a heavy-tailed distribution, which controls the variance of its target around the mean $\mathbf{x}^\top \boldsymbol{\theta}^*$. Training

observes one realization $y \sim Y_{\mathsf{x}}^*$ per input and fits a linear head (ridge/OLS). At inference, we evaluate pass@$k$ with a perfect verifier by drawing $k$ fresh realizations per test input and asking whether at least one lies within a fixed tolerance of the model prediction.

**Overview and contributions.**

- **A solvable LID model for linear fine-tuning.** We formalize the Latent Instance Difficulty (LID) model in the high-dimensional linear setting. Training with a single realization per input reduces to ridge/OLS and recovers established generalization scaling with respect to sample size $N$, dimension $d$, and spectral exponent $\alpha$ (including the $1/N$ tail in the classical regime when the average target variance is finite, i.e., $\beta > 2$).

- **Training–inference coupling via a two-tail law.** We show that the distribution of *single-trial success probabilities* under pass@$k$ acquires two regularly varying components: an *intrinsic* tail determined by the latent difficulty distribution (exponent $\beta$) and a *finite-$N$* tail governed by the model's error relative to the mean target (exponent $\gamma(N) \propto 1/\mathcal{L}_{\text{gen}}(N)$). Averaging over trials yields a mixture pass@$k$ law from which the *effective* inference exponent $\beta_{\text{eff}}(N) = \min\{\beta, \gamma(N)\}$ emerges. Hence the observed pass@$k$ slope *increases with $N$* and *saturates* at the intrinsic difficulty index $\beta$.

- **Predictions and implications.** The theory predicts (i) a crossover surface in $(N, k)$ separating a finite-$N$ (bias-dominated) region from an intrinsic-tail region; (ii) a saturating $\beta_{\text{eff}}(N)$ curve; and (iii) continued prefactor improvements with $N$ even after the slope has plateaued. These lead to a simple compute-allocation rule: invest in training until $\beta_{\text{eff}}(N)$ is near $\beta$, then prioritize inference attempts.

- **Evidence in simulation and a real-data proxy.** Controlled simulations confirm the $1/N$ training tail (with the correct intercept), the steepening of pass@$k$ with $N$, and a saturating $\hat{\beta}_{\text{eff}}(N)$. Two real-data proxies exhibit analogous behavior: CIFAR-10H (human-label variance) and a mathematical reasoning teacher–student distillation setting using GSM8K.

Taken together, these results provide a clean, testable baseline that unifies training and inference scaling in a setting that captures a widely used fine-tuning regime. The model makes explicit *when* test-time compute should help, *how far* its benefits can go, and *how* those benefits depend on training. In the LID linear fine-tuning setting, standard $1/N$-type generalization scaling with $N$ coexists with a heavy-tailed distribution of instance difficulty. As $N$ grows, the model's average squared bias shrinks and the error mass concentrates on a shrinking set of intrinsically hard examples. On the inference side, this shows up as a pass@$k$ failure curve $\mathcal{L}_{\text{inf}}(k; N) \sim \tilde{P}(N) k^{-\beta_{\text{eff}}(N)}$ whose slope $\beta_{\text{eff}}(N)$ increases with $N$ and eventually plateaus at the intrinsic difficulty index $\beta$. Equivalently, training progressively "shrinks" the hard tail of the error distribution until irreducible instance stochasticity dominates and the test time scaling exponent saturates.

## 2 RELATED WORK

Here, we provide a focused review of the prior research most central to our contribution. For an additional detailed literature survey, we refer the reader to App. A and references therein.

**Generalization Scaling Laws.** A large body of work has established that the generalization loss of deep networks often scales as a predictable power law with resources like dataset size $N$ or model parameters (Hestness et al., 2017; Kaplan et al., 2020; Hoffmann and et al., 2022). Theoretical frameworks seek to explain these laws by appealing to properties of the data, such as its spectral decay, or the model architecture (Bahri et al., 2021; Maloney et al., 2022). Our work builds on the standard scaling of generalization loss with $N$, which serves as the "training" component of our unified model.

**Inference-Time Scaling.** A parallel line of work has shown that performance on difficult reasoning tasks can be dramatically improved by increasing compute at inference time, even without further training (Snell et al., 2024; Brown et al., 2024). The dominant methods involve generating multiple candidate solutions and selecting the best one, often evaluated with the pass@$k$ metric. While some theoretical models for this phenomenon have been proposed (Levi, 2024; Schaeffer et al., 2025a), they typically analyze inference in isolation.

To our knowledge, a simple, solvable model that analytically connects the progress of training (i.e., the decrease in generalization error) to the scaling of inference performance is still missing. Our work aims to provide exactly this unified view.

The rest of the paper is organized as follows: In Sec. 3, we present the LID model. We provide our main results for training and inference scaling laws in Sec. 4. In Sec. 5 we present two examples of linear fine-tuning performed on an inherently stochastically labeled dataset, showing that the LID is a reasonable proxy for some real-world tasks. We conclude in Sec. 6.

## 3   THE LATENT INSTANCE DIFFICULTY SETTING

Real-world datasets exhibit significant instance-level heterogeneity: some image labels are ambiguous, incurring higher annotator disagreement (Peterson et al., 2019; Northcutt et al., 2021), and some reasoning problems are intrinsically harder than others, leading to more variable outputs. Standard homogeneous-noise assumptions overlook this complexity. Addressing this, and connecting it to the distinct scaling behaviors observed during training versus inference (especially when multiple inference attempts can be verified against a correct solution; cf. Section 1), motivates our setting. Intuitively, factors that increase training difficulty (harder to learn the mean) also shape inference reliability (harder to match a fresh realization).

**Last-layer fine-tuning view.** We instantiate *Latent Instance Difficulty* (LID) within the common fine-tuning regime: a frozen representation produces features $\mathbf{x} \in \mathbb{R}^d$ and we learn a linear head $\mathbf{x}^\top \boldsymbol{\theta}$. Each instance carries a latent *precision* (its "easiness") $\tau_\mathbf{x}$ that controls the variance of its target around the mean $\mathbf{x}^\top \boldsymbol{\theta}^*$. Training observes *one* realization $y$ per $\mathbf{x}$; at inference, pass@$k$ compares the model's prediction to $k$ fresh realizations from the same instance-specific target distribution.

**Definition 3.1** (Latent Instance Difficulty (LID) Model). The data generation process for an observation $(\mathbf{x}, y)$ is:

1. **Features.** An input feature vector $\mathbf{x} \in \mathbb{R}^d$ is drawn from a distribution $p(\mathbf{x})$ with zero mean $\mathbb{E}[\mathbf{x}] = 0$ and covariance $\boldsymbol{\Sigma} = \mathbb{E}[\mathbf{x}\mathbf{x}^T]$. We assume eigenvalues $\sigma_j^2$ exhibit power-law decay $\sigma_j^2 \propto j^{-(1+\alpha)}$ for $j = 1, \ldots, d$, with $\alpha > 0$.

2. **Latent difficulty.** Associated with $\mathbf{x}$ is a latent *difficulty precision* $\tau_\mathbf{x} \in (0, \infty)$, drawn independently of $\mathbf{x}$ from

$$\tau_\mathbf{x} \sim \mathrm{Gamma}(\mathrm{shape} = \beta/2, \ \mathrm{rate} = 1), \tag{1}$$

where $\beta > 0$ controls the near-zero tail. Smaller $\beta$ increases the mass at very low precision (high intrinsic variance), corresponding to "hard" instances.

3. **Stochastic target.** Conditional on $(\mathbf{x}, \tau_\mathbf{x})$, the instance target is Gaussian around the mean relationship $f^*(\mathbf{x}) = \mathbf{x}^\top \boldsymbol{\theta}^*$ with variance inversely proportional to $\tau_\mathbf{x}$:

$$Y_\mathbf{x}^* \sim \mathcal{N}\big(\mathrm{mean} = \mathbf{x}^T \boldsymbol{\theta}^*, \ \mathrm{variance} = \sigma_\eta^2/\tau_\mathbf{x}\big), \quad \text{where } \sigma_\eta^2 > 0 \text{ is a global scale.} \tag{2}$$

4. **Training labels.** The observed label is a single realization

$$y \sim Y_\mathbf{x}^* \quad \text{equivalently} \quad y = \mathbf{x}^T \boldsymbol{\theta}^* + \eta, \ \ \eta \sim \mathcal{N}(0, \ \sigma_\eta^2/\tau_\mathbf{x}). \tag{3}$$

**Comments.** (i) The power-law spectrum in Item 1 abstracts benign-feature regimes observed in practice and used in linear scaling analyses (Maloney et al., 2022; Levi and Oz, 2023; 2024). (ii) The Gamma family in Eq. (1) is chosen for analytic convenience; all results that govern inference scaling depend only on the *near-zero* tail $\Pr(\tau_\mathbf{x} \leq t) \asymp t^{\beta/2}$, so other distributions with the same tail index yield the same exponents (Levi, 2024). (iii) Independence of $\tau_\mathbf{x}$ and $\mathbf{x}$ simplifies exposition; allowing correlation is a natural extension and would primarily affect constants and crossover locations rather than exponents. (iv) We will denote the model's *bias relative to the mean target* on a fresh test feature by $\mathcal{E}_{\mathrm{gen}}(\mathbf{x}) := \mathbf{x}^\top \hat{\boldsymbol{\theta}} - \mathbf{x}^\top \boldsymbol{\theta}^*$; its distribution is governed by the training procedure and sample size $N$ (see Sec. 4).

**Corollary 3.2.** *The average variance of the target around its mean is*

$$\mathbb{E}\big[(Y_\mathbf{x}^* - \mathbf{x}^T \boldsymbol{\theta}^*)^2\big] = \mathbb{E}_\mathbf{x}[\mathrm{Var}(Y_\mathbf{x}^* \mid \mathbf{x})] = \sigma_\eta^2 \, \mathbb{E}\left[\frac{1}{\tau_\mathbf{x}}\right]. \tag{4}$$

*For* $\tau_{\mathbf{x}} \sim \text{Gamma}(\beta/2, 1)$, $\mathbb{E}[1/\tau_{\mathbf{x}}] = \frac{\Gamma(\beta/2-1)}{\Gamma(\beta/2)} = \frac{2}{\beta-2}$ *when* $\beta > 2$.

**Assumption 3.3** (Finite average target variance). We assume $\mathbb{E}[(Y_{\mathbf{x}}^* - \mathbf{x}^T\boldsymbol{\theta}^*)^2] < \infty$, i.e., $\beta > 2$.

Assumption 3.3 enables standard high-dimensional ridge/OLS analyses of $\mathcal{L}_{\text{gen}}$ (training/generalization scaling). Importantly, our inference-time results (pass@$k$ scaling) rely only on the small-$\tau$ tail and continue to hold in the sense of exponents even when $\beta \leq 2$ (though constants and the onset of asymptotics may change), provided the learned predictor is consistent.

## 3.1 TRAINING SETUP

We consider learning the mean relationship $f^*(\mathbf{x}) = \mathbf{x}^\top\boldsymbol{\theta}^*$ from a dataset $\mathcal{D}_N = \{(\mathbf{x}_i, y_i)\}_{i=1}^N$, where each $y_i$ is a *single* realization sampled according to Def. 3.1. In the fine-tuning view, $\mathbf{x}_i$ are frozen features from a pretrained backbone and we train only a linear head. The learner observes $(\mathbf{x}_i, y_i)$ and estimates $\hat{\boldsymbol{\theta}}$ by minimizing a ridge objective against the *realized* labels while evaluation is always against the *mean* target:

$$\mathcal{L}_{\text{train}}(\hat{\boldsymbol{\theta}}) = \frac{1}{N}\sum_{i=1}^N (y_i - \mathbf{x}_i^\top\hat{\boldsymbol{\theta}})^2 + \lambda\|\hat{\boldsymbol{\theta}}\|_2^2, \tag{5}$$

where $\hat{\boldsymbol{\theta}} \in \mathbb{R}^d$ (last-layer parameters) and $\lambda \geq 0$ is a small regularizer.

The minimizer admits the standard closed form

$$\hat{\boldsymbol{\theta}}_\lambda = \arg\min_{\hat{\boldsymbol{\theta}}} \mathcal{L}_{\text{train}}(\hat{\boldsymbol{\theta}}) = (N^{-1}X^\top X + \lambda I_d)^{-1} N^{-1}X^\top \mathbf{y}, \tag{6}$$

where $X \in \mathbb{R}^{N \times d}$ stacks the features and $\mathbf{y} \in \mathbb{R}^N$ the realized labels from equation 3. We will take the ridgeless limit $\lambda \to 0$ when well-posed; in the overparameterized case $N < d$ we use the standard scaling $\lambda = \tilde{\lambda}/N$ to recover the minimum-norm interpolator (see Sec. 4 and App.C).

**Bias relative to the mean target.** Throughout, we evaluate generalization against the *mean* signal $\mathbf{x}^\top\boldsymbol{\theta}^*$, and denote the model's instancewise deviation by

$$\mathcal{E}_{\text{gen}}(\mathbf{x}) := \mathbf{x}^\top\hat{\boldsymbol{\theta}}_\lambda - \mathbf{x}^\top\boldsymbol{\theta}^*. \tag{7}$$

The test (generalization) loss is $\mathcal{L}_{\text{gen}}(N, \lambda) = \mathbb{E}_{\mathbf{x}, \mathcal{D}_N}[\mathcal{E}_{\text{gen}}(\mathbf{x})^2]$. Under Ass. 3.3, the effective training noise has finite average variance $\sigma_{\text{noise}}^2 = \sigma_\eta^2 \mathbb{E}[1/\tau_{\mathbf{x}}] = \frac{2\sigma_\eta^2}{\beta-2}$, so standard high-dimensional ridge/OLS tools apply. In Sec. 4 we analyze how $\mathcal{L}_{\text{gen}}$ scales with $N$, $d$, and the feature spectrum exponent $\alpha$, and how the resulting distribution of $\mathcal{E}_{\text{gen}}(\mathbf{x})$ controls the finite-$N$ inference behavior (pass@$k$) and the effective inference exponent $\beta_{\text{eff}}(N)$.

## 4 THEORETICAL ANALYSIS OF SCALING LAWS

We analyze two coupled laws: (i) the dependence of the generalization loss $\mathcal{L}_{\text{gen}}$ on sample size $N$, and (ii) the pass@$k$ inference failure $\mathcal{L}_{\text{inf}}(k; N)$ on the number of trials $k$. In our fine-tuning view (last-layer regression on frozen features), $\mathcal{L}_{\text{gen}}$ controls the distribution of the instancewise deviation

$$\mathcal{E}_{\text{gen}}(\mathbf{x}) := \mathbf{x}^\top\hat{\boldsymbol{\theta}}_\lambda - \mathbf{x}^\top\boldsymbol{\theta}^*,$$

which in turn governs the finite-$N$ behavior of $\mathcal{L}_{\text{inf}}(k; N)$. As $N$ grows, $\mathcal{L}_{\text{gen}}(N)$ shrinks and the $\mathcal{E}_{\text{gen}}(\mathbf{x})$-induced penalty in $\mathcal{L}_{\text{inf}}(k; N)$ recedes; the observed inference slope transitions from a finite-$N$ regime to the latent-difficulty asymptote $-\beta$. We quantify this transition empirically via $\beta_{\text{eff}}(N)$ (Fig. 1, right).

### 4.1 TRAINING SCALING LAW $\mathcal{L}_{\text{GEN}}$ VS. $N$

We evaluate generalization against the mean target, so the test loss is

$$\mathcal{L}_{\text{gen}}(N, \lambda) = \mathbb{E}_{\mathbf{x}\sim p(\mathbf{x}), \mathcal{D}_N}\left[\left(\mathbf{x}^\top\hat{\boldsymbol{\theta}}_\lambda - \mathbf{x}^\top\boldsymbol{\theta}^*\right)^2\right]. \tag{8}$$

In high dimensions, $\mathcal{L}_{\text{gen}}$ depends critically on the $N$–$d$ ratio and on the spectrum of $\boldsymbol{\Sigma}$; we follow the standard decomposition into regimes (Belkin et al., 2019; Hastie et al., 2020). Under Assumption 3.3,

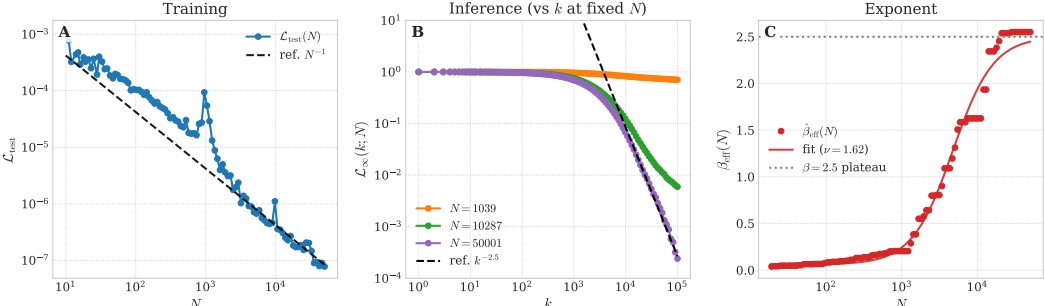

Figure 1: **Training and inference-time scaling laws in the LID setting.** *Left:* Generalization error $\mathcal{L}_{\text{gen}}$ vs. $N$, showing double descent and the two classical regimes. *Center:* Pass@$k$ inference failure rate $\mathcal{L}_{\text{inf}}(k; N)$ vs. $k$ for several $N > d$, with asymptotic slope $-\beta$ (dashed black) once the mean is well learned. *Right:* The effective inference exponent $\beta_{\text{eff}}(N)$ extracted from the local log–log slope in a fixed $k$-window. The dotted line marks the asymptote $\beta$; the solid curve is the empirical fit $\beta_{\text{eff}}(N) = \beta - \Delta/(1 + c_\beta N^\nu)$. We use $\lambda = 10^{-9}$ and $\sigma_\eta = 10^{-3}$.

the effective label noise has finite average variance $\sigma_{\text{noise}}^2 = \sigma_\eta^2 \mathbb{E}[1/\tau_{\mathbf{x}}]$, and classical ridge/OLS results apply (see App. C for a concise derivation in our notation).

**Overparameterized regime** ($N < d$). With ridgeless (or lightly regularized) interpolation, the error is governed by the minimum-norm bias and the data spectrum exponent $\alpha$ (Bartlett et al., 2020; Mei and Montanari, 2020; Hastie et al., 2020; Maloney et al., 2022). For power-law spectra $\sigma_j^2 \propto j^{-(1+\alpha)}$, one obtains

$$\mathcal{L}_{\text{gen}}(N) \;\propto\; P_N \, N^{-\alpha}, \qquad N < d, \tag{9}$$

where $P_N$ is a benign prefactor encapsulating spectrum and teacher alignment. The symbol $\propto$ here means that $\mathcal{L}_{\text{gen}}(N)/\big(N^{-\alpha}\big)$ remains bounded away from zero and infinity as $N \to \infty$ along this regime (cf. Section B).

**Underparameterized regime** ($N > d$). When samples exceed parameters, the variance term dominates and decays at the parametric rate,

$$\mathcal{L}_{\text{gen}}(N) \;\propto\; \sigma_\eta^2 \, \mathbb{E}\!\left[\frac{1}{\tau_{\mathbf{x}}}\right] \frac{d}{N}, \qquad N \gg d, \tag{10}$$

with the usual $1/N$ slope and a noise-controlled constant reflecting that the mean target is learned from single-shot labels. The precise constant can be expressed in terms of the inverse Marchenko–Pastur law when $\mathbf{x}$ are Gaussian with covariance $\boldsymbol{\Sigma}$ (App. C).

**Transition** ($N \approx d$). Near interpolation the estimator is ill-conditioned and $\mathcal{L}_{\text{gen}}$ exhibits a peak ("double descent"), whose height and width depend on $\lambda$ and on the spectrum (Belkin et al., 2019).

In the overparameterized regime, performance is limited by how quickly the model can resolve low-variance spectral modes of the data, leading to the $N^{-\alpha}$ tail. Once the model has more samples than parameters, the effective noise from single-shot labels dominates and $\mathcal{L}_{\text{gen}}(N)$ becomes variance-limited, decaying as $d/N$. This classical behavior provides the backdrop against which we interpret the training-dependent changes in inference scaling.

The left panel of Fig. 1 confirms these scalings in our LID setting and provides the $\mathcal{L}_{\text{gen}}(N)$ baselines used to interpret the finite-$N$ inference behavior discussed next.

## 4.2 INFERENCE SCALING LAW $\mathcal{L}_{\text{INF}}$ VS. $k$

We analyze inference through the pass@$k$ metric (Snell et al., 2024; Brown et al., 2024), i.e., the probability that at least one of $k$ independent trials matches the target within a fixed tolerance. In the LID setting, for a test instance $\mathbf{x}$ with latent precision $\tau_{\mathbf{x}}$,

$$Y_{\mathbf{x}}^* \sim \mathcal{N}\big(\mathbf{x}^\top \boldsymbol{\theta}^*, \, \sigma_\eta^2/\tau_{\mathbf{x}}\big),$$

and the model outputs $\hat{y} = \mathbf{x}^\top \hat{\boldsymbol{\theta}}_\lambda$. On trial $j$ we draw $y_j \sim Y_{\mathbf{x}}^*$, and the error is

$$e_j \coloneqq \hat{y} - y_j = \big(\mathbf{x}^\top \hat{\boldsymbol{\theta}}_\lambda - \mathbf{x}^\top \boldsymbol{\theta}^*\big) - \eta_j = \mathcal{E}_{\text{gen}}(\mathbf{x}) - \eta_j, \qquad \eta_j \sim \mathcal{N}\big(0, \, \sigma_\eta^2/\tau_{\mathbf{x}}\big).$$

A trial succeeds if $|e_j| \leq \delta$ for a fixed tolerance $\delta > 0$.

**Assumption 4.1** (Perfect verification). We assume a perfect verifier that declares overall success for a given $\mathbf{x}$ iff at least one of the $k$ trials satisfies $|e_j| \leq \delta$.

Let $p(\mathbf{x}, \tau_{\mathbf{x}})$ denote the single-trial failure probability

$$p(\mathbf{x}, \tau_{\mathbf{x}}) := \mathbb{P}_{\eta \sim \mathcal{N}(0, \sigma_\eta^2 / \tau_{\mathbf{x}})}\big(|\mathcal{E}_{\text{gen}}(\mathbf{x}) - \eta| > \delta\big).$$

Assuming conditional independence across trials, the pass@$k$ failure probability is

$$\mathcal{L}_{\text{inf}}(k; N) := \mathbb{E}_{\mathbf{x}, \tau_{\mathbf{x}}, \mathcal{D}_N}\big[p(\mathbf{x}, \tau_{\mathbf{x}})^k\big] = 1 - \text{pass@}k. \tag{11}$$

**Remark (model-draw vs. target-draw pass@$k$).** In practical pass@$k$ for Large Language Models (LLMs) one draws $k$ *model* outputs and verifies them against a fixed target; here we draw $k$ *target* realizations and compare them to a fixed predictor $\hat{y} = \mathbf{x}^\top \hat{\boldsymbol{\theta}}_\lambda$. Under a small tolerance window and smooth local densities, the single-try success probability is proportional to a local density evaluated at the prediction gap $B_N(\mathbf{x}) := \mathcal{E}_{\text{gen}}(\mathbf{x})$. Whether the randomness is in the model or in the target then affects only constants, not the asymptotic exponent of the pass@$k$ curve; we formalize this equivalence in App. D.2.

**Bias-free asymptotic tail.** We first consider the asymptotic scaling when the mean is learned well enough that the instancewise bias $B_N(\mathbf{x})$ is negligible compared to $\sigma_\eta / \sqrt{\tau_{\mathbf{x}}}$ for the small-$\tau_{\mathbf{x}}$ instances that dominate the failures. In that regime, a small-window expansion in $\delta$ yields

$$1 - p(\mathbf{x}, \tau_{\mathbf{x}}) = c_\delta \sqrt{\tau_{\mathbf{x}}}\big(1 + o(1)\big) \quad \text{as } \delta \downarrow 0,$$

for a constant $c_\delta = 2\delta/(\sqrt{2\pi}\,\sigma_\eta)$; see App. D for details. A Tauberian analysis of the Laplace–Stieltjes transform under the Gamma prior $\tau_{\mathbf{x}} \sim \text{Gamma}(\beta/2, 1)$ then gives the following formal statement.

**Proposition 4.2** (Bias-free pass@$k$ scaling). *Suppose Assumption 3.3 and the regular-variation and smoothness conditions in App. E.2 hold, and consider the limiting bias-free predictor (or a sequence of predictors for which $B_N(\mathbf{x}) \to 0$ in distribution). Then, as $k \to \infty$,*

$$\mathcal{L}_{\text{inf}}(k) = \tilde{P}_{\beta, \delta, \sigma_\eta}\, k^{-\beta}\big(1 + o(1)\big), \qquad \tilde{P}_{\beta, \delta, \sigma_\eta} = \frac{2\,\Gamma(\beta)}{\Gamma(\beta/2)\, c_\delta^\beta}, \tag{12}$$

*where $c_\delta = 2\delta/(\sqrt{2\pi}\,\sigma_\eta)$.*

Thus the asymptotic slope $-\beta$ is governed entirely by the small-$\tau_{\mathbf{x}}$ tail of the LID prior and is independent of $d/N$.

**Finite-$N$ correction and training-dependent exponent.**

For finite $N$, the instancewise bias $B_N(\mathbf{x}) = \mathcal{E}_{\text{gen}}(\mathbf{x})$ is non-negligible and suppresses the per-trial success probability at moderate $\tau_{\mathbf{x}}$. For small $\delta$, a Gaussian CDF expansion yields

$$1 - p(\mathbf{x}, \tau_{\mathbf{x}}) = \frac{\sqrt{2}}{\sqrt{\pi}} \frac{\delta}{\sigma_\eta} \sqrt{\tau_{\mathbf{x}}} \exp\Big(-\frac{B_N(\mathbf{x})^2\, \tau_{\mathbf{x}}}{2\sigma_\eta^2}\Big)\big(1 + o(1)\big), \tag{13}$$

uniformly for $\tau_{\mathbf{x}}$ in compact subsets of $(0, \infty)$ as $\delta \downarrow 0$; see App. E.2. Under standard linear generalization, $B_N(\mathbf{x})$ is approximately Gaussian with $\mathbb{E}[B_N(\mathbf{x})] = 0$ and

$$\text{Var}[B_N(\mathbf{x})] = \Theta\big(\mathcal{L}_{\text{gen}}(N)\big).$$

Using equation 13 and Tauberian arguments for Laplace–Stieltjes transforms (detailed in App. E), we obtain a two-tail mixture law for $\mathcal{L}_{\text{inf}}(k; N)$.

**Theorem 4.3** (Two-tail mixture law for pass@$k$). *Let the assumptions in App. E.2 hold, and suppose $B_N(\mathbf{x})$ is centered sub-Gaussian with $\text{Var}[B_N(\mathbf{x})] \asymp \mathcal{L}_{\text{gen}}(N)$ and independent of $\tau_{\mathbf{x}}$. Then there exist positive constants $\tilde{P}$ and $\tilde{P}_N(N)$ and a function $\gamma(N) > 0$ such that, as $k \to \infty$,*

$$\mathcal{L}_{\text{inf}}(k; N) = \tilde{P}\, k^{-\beta} + \tilde{P}_N(N)\, k^{-\gamma(N)}\big(1 + o(1)\big), \tag{14}$$

*with*

$$\gamma(N) = \Theta\Big(\frac{1}{\text{Var}[B_N(\mathbf{x})]}\Big) = \Theta\Big(\frac{1}{\mathcal{L}_{\text{gen}}(N)}\Big). \tag{15}$$

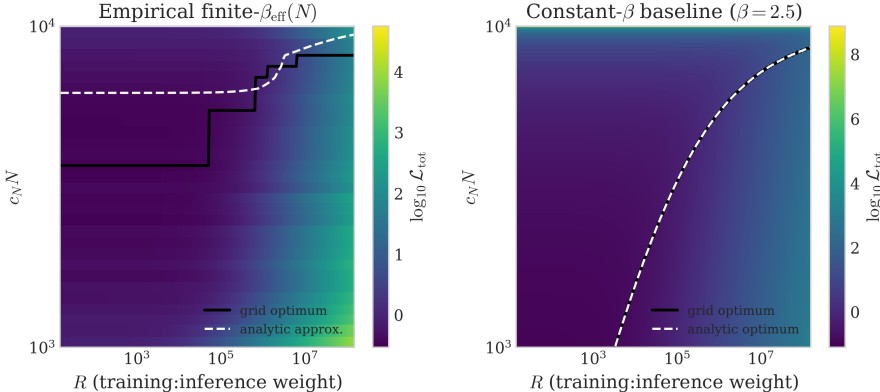

Figure 2: **Compute allocation with a training-dependent inference exponent.** *Left:* empirical finite-$N$ case. We plot $\log_{10} \mathcal{L}_{\text{tot}}(N, k; R)$ with $R$ the training:inference weight ratio and $C = Nc_N + kc_k$. Black: grid optimum $\tilde{N}_\star$; white dashed: analytic approximation from equation 21, using $\beta_{\text{eff}}(N)$ (local log–log slope of $\mathcal{L}_{\text{inf}}$) and its discrete derivative. *Right:* constant-$\beta$ baseline calibrated from the same data; white dashed is the closed-form analytic optimum. Contours clearly shift toward larger $N$ in the finite-$N$ panel, consistent with the $-\beta'_{\text{eff}}(N) \log(\cdot)$ correction.

The first term in equation 14 is the intrinsic LID tail governed by $\beta$; the second term is a finite-$N$ correction arising from large bias $|B_N(\mathbf{x})|$ at moderate $\tau_{\mathbf{x}}$.

**Corollary 4.4** (Training-dependent effective exponent). *Fix a $k$-window $[k_1, k_2]$ such that for each $N$ either the $k^{-\beta}$ or the $k^{-\gamma(N)}$ term in equation 14 dominates over that window. Then, as $k_1 \to \infty$,*

$$\beta_{\text{eff}}(N; [k_1, k_2]) = \min\{\beta, \gamma(N)\} + o(1), \tag{16}$$

*where $\beta_{\text{eff}}(N; [k_1, k_2])$ is defined in equation 23. In particular, as $N$ increases and $\mathcal{L}_{gen}(N)$ decreases, $\gamma(N)$ increases and the effective inference exponent $\beta_{\text{eff}}(N)$ monotonically approaches the intrinsic LID exponent $\beta$.*

The two exponents $\beta$ and $\gamma(N)$ play distinct roles. The intrinsic exponent $\beta$ comes solely from the heavy tail of the instance difficulty distribution and cannot be improved by additional training once the mean target has been learned. By contrast, $\gamma(N)$ encodes a finite-$N$ penalty: when the model's mean error is still large, many test points behave as if they were "effectively hard" even if their latent $\tau_{\mathbf{x}}$ is moderate. As training drives down $\mathcal{L}_{\text{gen}}(N)$, this artificial hard tail disappears, and the observed pass@$k$ curves steepen until $\beta_{\text{eff}}(N)$ is limited only by $\beta$.

In practice we summarize this monotone saturation by the empirical fit

$$\beta_{\text{eff}}(N) = \beta - \frac{\Delta}{1 + c_\beta N^\nu}, \tag{17}$$

where $(\Delta, c_\beta, \nu)$ depend weakly on the $k$-window and on the estimator. The center and right panels of Fig. 1 illustrate: (i) for fixed $N > d$, $\mathcal{L}_{\text{inf}}(k; N)$ exhibits a slope that steepens with $N$; (ii) $\beta_{\text{eff}}(N)$ increases with $N$ and plateaus at $\beta$.

## 4.3 COMPUTE ALLOCATION TRADEOFF WITH TRAINING-DEPENDENT $\beta_{\text{eff}}(N)$

Having characterized the finite-$N$ inference scaling, we can combine training and inference laws to study the optimal budget allocation between the two.

We consider a fixed compute budget $C$ that must be split between training samples (cost $c_N$ each) and inference trials (cost $c_k$ each) $C = Nc_N + kc_k$, s.t. $k = \frac{C - \tilde{N}}{c_k}, \tilde{N} := Nc_N$. We minimize a weighted objective

$$\mathcal{L}_{\text{tot}}(\tilde{N}) = R\,\mathcal{L}_{\text{gen}}(N) + \mathcal{L}_{\text{inf}}\Big(k = \frac{C - \tilde{N}}{c_k}; N\Big), \qquad \text{training/inference weight} = R > 0. \tag{18}$$

In the classical (under-parameterized) regime $N \gg d$ we have $\mathcal{L}_{\text{gen}}(N) \propto P_N N^{-\gamma}$ with $\gamma = 1$ (Sec. 4.1); more generally one may take $\gamma \in \{1, \alpha\}$ depending on $d/N$. For inference, Theorem 4.3

suggests modeling $\mathcal{L}_{\inf}(k; N)$ over the practical $k$-window by

$$\mathcal{L}_{\inf}(k; N) \approx \tilde{P}(N) \, k^{-\beta_{\mathrm{eff}}(N)}, \qquad \beta_{\mathrm{eff}}(N) \uparrow \beta \text{ as } N \to \infty, \tag{19}$$

where $\tilde{P}(N)$ is a slowly varying prefactor capturing residual bias effects. Here "$\approx$" is to be understood in the sense of an empirical fit over a fixed $k$-window, not as an asymptotic statement; the asymptotic behavior as $k \to \infty$ is given by Theorem 4.3.

With $\tilde{N} = Nc_N$ and $k = (C - \tilde{N})/c_k$, the budget-constrained objective becomes

$$\mathcal{L}_{\mathrm{tot}}(\tilde{N}) = R \, P_N \, c_N^\gamma \, \tilde{N}^{-\gamma} + \tilde{P}(N) \, c_k^{\beta_{\mathrm{eff}}(N)} \, (C - \tilde{N})^{-\beta_{\mathrm{eff}}(N)}. \tag{20}$$

**Proposition 4.5** (Optimal allocation with training-dependent $\beta_{\mathrm{eff}}(N)$)**.** *Assume $\mathcal{L}_{gen}(N) \propto P_N N^{-\gamma}$ and that over the practical $k$-window the pass@k failure satisfies $\mathcal{L}_{inf}(k; N) \approx \tilde{P}(N)k^{-\beta_{\mathrm{eff}}(N)}$ with $\beta_{\mathrm{eff}}(N)$ differentiable in $N$. Let $N = \tilde{N}/c_N$. Any interior optimum $\tilde{N}_* \in (0, C)$ of $\mathcal{L}_{\mathrm{tot}}(\tilde{N})$ satisfies*

$$R \, P_N \, c_N^\gamma \, \gamma \, \tilde{N}^{-(\gamma+1)} = \tilde{P}(N) \, c_k^{\beta_{\mathrm{eff}}(N)} \, (C - \tilde{N})^{-\beta_{\mathrm{eff}}(N)} \left[ \frac{\beta_{\mathrm{eff}}(N)}{C - \tilde{N}} - \frac{\beta'_{\mathrm{eff}}(N)}{c_N} \ln\left(\frac{C - \tilde{N}}{c_k}\right) \right]. \tag{21}$$

*If $\tilde{P}(N)$ is slowly varying and $|\beta'_{\mathrm{eff}}(N)|$ is small over the relevant range, equation 21 simplifies to the quasi-static balance*

$$R \, P_N \, c_N^\gamma \, \gamma \, \tilde{N}^{-(\gamma+1)} \approx \tilde{P} \, c_k^\beta \, \beta \, (C - \tilde{N})^{-(\beta+1)}, \tag{22}$$

*where $\tilde{P}$ is an $N$-independent constant and the approximation is understood in the sense that the two sides match up to slowly varying multiplicative factors.*

Compared to the constant-$\beta$ condition, equation 21 includes a *new logarithmic term* proportional to $-\beta'_{\mathrm{eff}}(N) \ln\big((C - \tilde{N})/c_k\big)$. When the budget is such that $k = (C - \tilde{N})/c_k$ lies below the LID-dominated window (so $\beta'_{\mathrm{eff}}(N) > 0$ and $\ln((C - \tilde{N})/c_k) > 0$), this term increases the marginal benefit of training, shifting the optimum towards larger $N$. Once $\beta_{\mathrm{eff}}(N)$ has saturated (so $\beta'_{\mathrm{eff}}(N) \approx 0$), equation 21 reduces to the constant-exponent balance in equation 22 (recovering the classical tradeoff with $\beta$ replaced by $\beta_{\mathrm{eff}}(N)$).

In particular:

- *Under-parameterized, near saturation.* With $\gamma = 1$, $\beta'_{\mathrm{eff}}(N) \approx 0$ and slowly varying $\tilde{P}(N)$, equation 22 gives an accurate allocation rule: allocate training until the marginal $N$-gain $\propto \tilde{N}^{-2}$ matches the marginal $k$-gain $\propto (C - \tilde{N})^{-(\beta_{\mathrm{eff}}+2)}$.

- *Finite-$N$, sub-asymptotic $k$.* When $\beta_{\mathrm{eff}}(N)$ is still increasing, the $-\beta'_{\mathrm{eff}}(N) \log(\cdot)$ term makes additional training strictly more valuable than the quasi-static approximation predicts; optimal policies invest more in $N$ until the effective slope stabilizes.

This shift toward larger $N$ in the finite-$N$ regime is evident in Fig. 2.

## 5 Empirical Results on Real World Applications

To bridge the LID model with realistic settings, we perform two experiments: we train a linear classifier on features extracted from CIFAR-10H, and perform a teacher–student distillation experiment on reasoning tasks using auto-regressive LLMs. We discuss the results of these experiments in the following sections, with additional details provided in Apps. F and H.

### 5.1 Test Case I: LID in CIFAR-10H with pre-trained features

Here, we validate the LID predictions on CIFAR-10 (Krizhevsky, 2012) paired with human label distributions from the annotated CIFAR-10H dataset (Peterson et al., 2019). We *freeze* a pretrained ResNet-18 (He et al., 2015) and *fine-tune* only a linear head on top of the backbone features, treating human disagreement as instance-dependent noise (high variance $\Rightarrow$ low precision), matching the LID setting. Full protocol and implementation details are in App. F.

Fig. 3 (top) shows $\mathcal{L}_{\mathrm{gen}}$ vs. $N$ and $\mathcal{L}_{\inf}(\cdot; N)$ vs. $k^1$. The generalization curve exhibits the familiar transition near $N \approx d$ and the $1/N$ decay for $N \gg d$, consistent with our linear-ridge analysis. For

---

[1]Here, $s = 32$ and $\delta = 0.05$ following the notations in App. F.

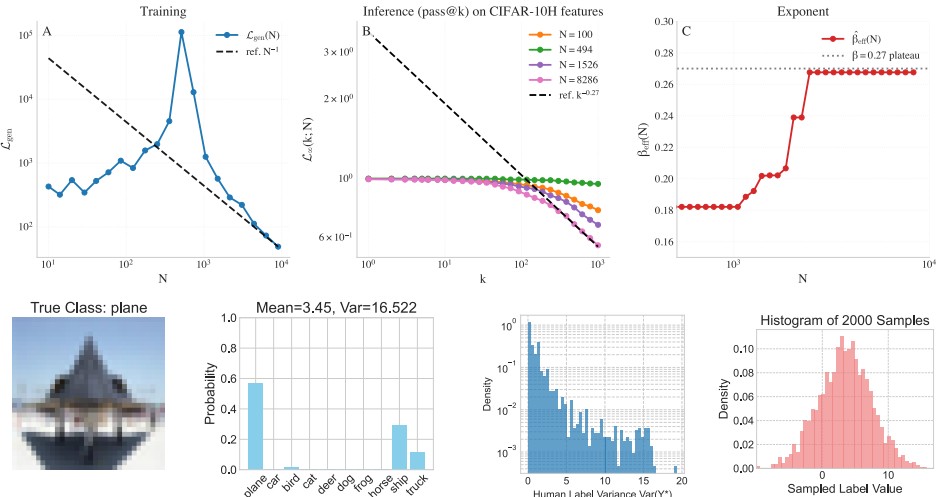

Figure 3: **Training and inference scaling on CIFAR-10H (frozen backbone, linear head). Top row:** *Left:* Generalization loss $\mathcal{L}_{\text{gen}}(N)$ with the $N^{-1}$ tail in the classical regime. *Center:* Pass@$k$ failure $\mathcal{L}_{\text{inf}}(k; N)$ for several $N$ on CIFAR-10H; dashed shows a $k^{-\beta}$ reference anchored on the largest-$N$ curve. *Right:* Effective inference slope $\beta_{\text{eff}}(N)$ estimated from the local log–log slope; the saturating fit $\beta_{\text{eff}}(N) = \beta - \Delta/(1 + c_\beta N^\nu)$ is overlaid, approaching the intrinsic tail index $\beta$. **Bottom row:** A CIFAR-10 image, its human label distribution, and the Gaussian PDF used to sample training/inference labels, illustrating instance difficulty via label variance.

inference, each fixed-$N$ curve follows an approximate power law $k^{-\beta_{\text{eff}}(N)}$; we estimate $\beta_{\text{eff}}(N)$ as the local log–log slope in a central $k$ window. As predicted by Sec. 4.2, $\beta_{\text{eff}}(N)$ *increases with* $N$ and *saturates*, reflecting the crossover from a bias-limited window (finite-$N$ mean error) to the intrinsic LID-dominated tail.

The bottom row illustrates how human disagreement induces instance difficulty: broad label histograms (large instance noise variance) produce wider Gaussians in equation 48, making pass@$k$ success rarer at small $k$. This matches the mechanism behind the $k^{-\beta}$ law in the synthetic LID model (Sec. 4.2), with the caveat that the empirical difficulty distribution is not exactly Gamma; nevertheless, the slope behavior is robust and governed by the prevalence of high-variance instances.

**Takeaways.** (i) The linear-head fine-tuning setting reproduces the $\mathcal{L}_{\text{gen}}$ scaling predicted by the ridge analysis. (ii) The pass@$k$ failure curves exhibit log–log linearity with a slope $\beta_{\text{eff}}(N)$ that grows with training data and plateaus, aligning with the finite-$N$ theory and supporting the compute-allocation results in Sec. 4.3. (iii) Treating human uncertainty as instance-dependent noise provides a realistic testbed for LID: the qualitative scaling and the training-dependent inference exponent persist despite deviations from the idealized Gamma prior or independence assumptions between $\tau_{\mathbf{x}}$ and $\mathbf{x}$.

## 5.2 TEST CASE II: GSM8K TEACHER–STUDENT DISTILLATION

We complement our CIFAR-10H experiment with a small GSM8K (Cobbe et al., 2021) distillation task designed to probe the LID training–inference coupling. A `Flan-T5-XL` (Chung et al., 2022) teacher produces solution traces on GSM8K-train; a `Flan-T5-small` student is fine-tuned with LoRA (Hu et al., 2021) ($r=8$) on subsets of sizes $N \in [10, 6309]$ (one teacher sample per question). We evaluate on GSM8K-test using two metrics: (i) a *strict greedy* proxy for training loss that decodes a single student answer, parses only explicit "final answer" numerics, and reports valid-only MSE (with $1\%$ Winsorization) at tolerance $\delta = 10^{-3}$; and (ii) the pass@$k$ failure $\mathcal{L}_{\text{inf}}(k; N)$ together with the effective slope $\hat{\beta}_{\text{eff}}(N)$ estimated from a central $k$-window.

Fig. 4 shows that the strict greedy loss decreases modestly with $N^2$, while $\mathcal{L}_{\text{inf}}(k; N)$ steepens and $\hat{\beta}_{\text{eff}}(N)$ increases then saturates, which is precisely the two-tail behavior predicted by the LID model (§4.2, App. E). Full setup and evaluation details appear in App. H.

---

[2]While the test loss is unstable due to large numerical mismatches (some final answers are simply larger numbers than others), the overall trend decreases and the inference metrics are stable.

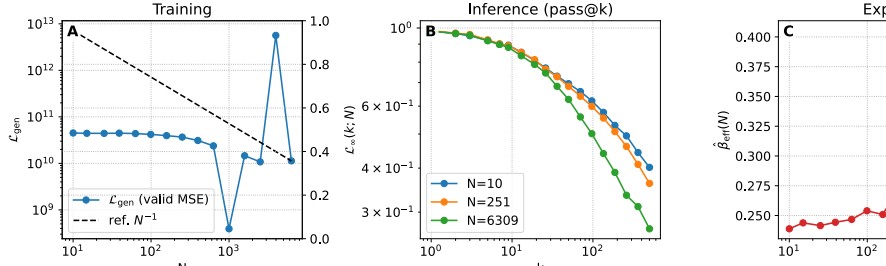

Figure 4: **GSM8K distillation (teacher→student).** *Left:* strict greedy valid-only MSE vs. $N$ (1% Winsorized), with a $N^{-1}$ reference. *Center:* $\mathcal{L}_{\text{inf}}(k; N)$ vs. $k$ for three $N$ values. *Right:* effective inference slope $\hat{\beta}_{\text{eff}}(N)$, which increases with $N$ and plateaus, consistent with the LID two-tail law.

## 6  DISCUSSION AND CONCLUSION

This work introduced the Latent Instance Difficulty (LID) model, a simple, solvable framework for last-layer fine-tuning that unifies the scaling laws of training and inference. We modeled tasks with intrinsic, instance-heterogeneous difficulty and showed that while the generalization loss, $\mathcal{L}_{\text{gen}}$, follows established scaling with sample size $N$ and data spectrum $\alpha$, its improvement has a direct and non-trivial impact on inference performance.

Our central contribution is the derivation of a **training-dependent inference exponent**. The pass@$k$ failure rate, $\mathcal{L}_{\text{inf}}(k)$, decays as a power law, but its exponent, $\beta_{\text{eff}}(N)$, is not fixed. It begins small for poorly-trained models and grows with the number of training samples $N$, eventually saturating at an intrinsic limit, $\beta$, determined by the tail of the task's true difficulty distribution. This mechanism reveals how reducing the model's error relative to the mean target makes inference-time compute more effective, up to a point of diminishing returns set by the data's irreducible stochasticity.

This unified view yields actionable insights. It predicts a clear crossover in the optimal resource allocation strategy: when the model is undertrained and $\beta_{\text{eff}}(N) < \beta$, the marginal benefit of acquiring more training data is high. Once the model is well-trained and the inference exponent has saturated, further gains are best sought by investing in more inference-time compute.

The LID model, while simple, provides a valuable theoretical baseline. It cleanly separates the roles of data structure ($\alpha$) and data heterogeneity ($\beta$) while also explaining how they are coupled through the training process. By providing a closed-form, testable theory for when and how much test-time compute should help, it offers a first step toward a more principled understanding of resource allocation in modern machine learning.

Although we instantiate LID in last-layer fine-tuning with scalar regression, the core mechanism: pass@$k$ failure mixes an *intrinsic* difficulty tail (index $\beta$) with a *training-dependent* tail controlled by $\mathcal{L}_{\text{gen}}(N)$, is model-agnostic. Beyond fine-tuning, training from scratch can be analyzed by allowing the representation to evolve, effectively changing the feature spectrum (e.g., the decay exponent $\alpha$) and thus the rate at which the finite-$N$ tail collapses; the saturation level may then be architecture-dependent. For classification, instance difficulty can be identified with label ambiguity/noise, and pass@$k$ is the event that at least one of $k$ draws hits the correct class, yielding the same two-tail mixture and a saturating $\beta_{\text{eff}}(N)$. In reinforcement learning, difficulty corresponds to return variance, so pass@$k$ is the chance that $k$ rollouts contain a near-optimal return; again the LID predictions carry over. For LLM reasoning, increasing pre-training tokens reduces mean error and empirically steepens the pass@$k$ slope; we conjecture $\beta_{\text{eff}}(N)$ grows with scale and saturates at an intrinsic difficulty index, with the same observable signature: as training improves, pass@$k$ curves steepen and then plateau.

**Limitations.** We analyze a linear last-layer model for clarity; extending the theory to multi-layer, non-linear architectures is an important next step, with random-feature or kernel models a natural bridge. We also treat the intrinsic difficulty distribution as task-fixed; for complex LLM reasoning, stronger models may both reduce bias and effectively "simplify" the task, shifting the apparent tail index $\beta$. Thus our results should be read as describing, for a *fixed* architecture, how $\beta_{\text{eff}}(N)$ rises toward an architecture-dependent plateau; quantifying how that plateau varies across models remains open. Finally, we study single-output regression; carrying the LID perspective on training–inference coupling to fully structured, auto-regressive outputs (as in LLMs) is a key challenge for future work.

## 7 Acknowledgements

We thank Yohai Bar-Sinai, Alon Beck, Francesco Cagnetta, Joshua Kazdan, Nadav Outmezguine and Zohar Ringel for fruitful discussions. NL is supported by the EPFL AI4science/AI Center program.

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

## A  RELATED WORK

**Generalization Scaling Laws** Neural scaling laws (NSL) have been shown to accurately describe how the generalization loss of deep neural networks improves with scale (model/dataset size, compute) (Hestness et al., 2017; Kaplan et al., 2020; Hoffmann and et al., 2022). Language model cross-entropy loss, for instance, often exhibits power-law scaling over orders of magnitude (Kaplan et al., 2020; Henighan et al., 2020), suggesting quantifiable performance gains with increased resources. Theoretical frameworks aim to explain these observations by identifying distinct scaling regimes (e.g., variance-limited, resolution-limited) (Bahri et al., 2021; Sharma and Kaplan, 2022; Spigler et al., 2019), sometimes linking them to data manifold intrinsic dimension (Sharma and Kaplan, 2022; Bahri et al., 2021). More nuanced models address dynamical scaling evolution with training time (Bordelon et al., 2024) and "broken" neural scaling laws (BNSL) that capture transitions between power-law regimes (Caballero et al., 2023; Nakkiran et al., 2021). Such breaks highlight complexities in extrapolation and suggest plateaus might stem from suboptimal strategies rather than fundamental limits (Sorscher et al., 2022; Dey et al., 2025). Beyond pre-training, scaling laws also govern fine-tuning performance relative to pre-trained model size and fine-tuning data volume (Hernandez et al., 2021; Lin et al., 2024). These studies show pre-training effectively augments fine-tuning data, with transfer benefits following predictable, though task-dependent, patterns (Hernandez et al., 2021), shifting focus towards leveraging pre-training for diverse downstream applications. In this work, we mainly focus on the vanilla NSL setting where the scaling law pertains to generalization loss improvement with increased number of training samples.

**Inference Time Scaling**

Methods for scaling inference-time compute in deep learning often involve generating multiple solution candidates and then selecting the best one based on specific criteria. These criteria include choosing the most frequent response for majority voting or the best response based on an external reward for Best-of-N (Brown et al., 2024; Irvine et al., 2023; Levi, 2024; Muennighoff et al., 2025; Schaeffer et al., 2025a; Kazdan et al., 2025; Schaeffer et al., 2025b; Chen et al., 2025). Unlike repeated sampling, previous sequential scaling methods let the model generate solution attempts sequentially, building upon previous attempts and allowing it to refine each attempt based on prior outcomes (Snell et al., 2024; Hou et al., 2025; Lee et al., 2025). Tree-based search methods (Gandhi et al., 2024; Wu et al., 2024) offer a hybrid approach between sequential and parallel scaling. Examples include Monte-Carlo Tree Search (MCTS) (Liu et al., 2024; Zhang et al., 2023; Zhou et al., 2024; Choi et al., 2023) and guided beam search (Xie et al., 2023). REBASE (Wu et al., 2024) employs a process reward model to balance exploitation and pruning during its tree search. Empirically, REBASE has been shown to outperform sampling-based methods and MCTS (Wu et al., 2024). Reward models play a key role in these inference-time scaling methods (Lightman et al., 2023; Wang et al., 2024a;b; Wu et al., 2024; Gandhi et al., 2024; Liu et al., 2024; Zhang et al., 2023; Zhou et al., 2024; Choi et al., 2023; Xie et al., 2023; Xin et al., 2024; Ankner et al., 2024). They generally come in two variants: outcome reward models and process reward models. Outcome reward models (Xin et al., 2024; Ankner et al., 2024) assign a score to complete solutions and are particularly useful in Best-of-N selection. In contrast, process reward models (Lightman et al., 2023; Wang et al., 2024a; Wu et al., 2024) assess individual reasoning steps and are effective in guiding tree-based search methods. The effect of reinforcement learning has recently been analyzed by Tsilivis et al. (2025). Other approaches also explore simple test-time scaling techniques (Muennighoff et al., 2025).

## B  NOTATION AND ASYMPTOTIC CONVENTIONS

We briefly summarize the main notation and asymptotic conventions used here and in the main body of the paper.

**Metrics.** The generalization (test) loss against the mean target is

$$\mathcal{L}_{\text{gen}}(N, \lambda) \coloneqq \mathbb{E}_{\mathbf{x}, \mathcal{D}_N}\left[\left(\mathbf{x}^\top \hat{\boldsymbol{\theta}}_\lambda - \mathbf{x}^\top \boldsymbol{\theta}^*\right)^2\right].$$

The pass@$k$ failure probability is

$$\mathcal{L}_{\text{inf}}(k; N) \coloneqq \mathbb{P}\Big(\text{at least } k \text{ trials all fail under the pass@}k \text{ protocol}\Big),$$

so that pass@$k = 1 - \mathcal{L}_{\text{inf}}(k; N)$.

**Asymptotic notation.** For sequences or functions $f$ and $g$ depending on a variable $t$ (typically $t = k$ or $t = N$), we write:

- $f(t) \sim g(t)$ as $t \to \infty$ if $\lim_{t \to \infty} f(t)/g(t) = 1$.
- $f(t) = O(g(t))$ as $t \to \infty$ if there exists $C > 0$ and $t_0$ such that $|f(t)| \le C|g(t)|$ for all $t \ge t_0$.
- $f(t) = \Theta(g(t))$ as $t \to \infty$ if $f(t) = O(g(t))$ and $g(t) = O(f(t))$.
- $f(t) \asymp g(t)$ if there exist constants $0 < c_1 \le c_2 < \infty$ such that $c_1 \le f(t)/g(t) \le c_2$ for all $t$ in the regime under consideration.

We use $o(1)$ for quantities that tend to $0$ in the relevant limit (which will always be explicitly indicated, e.g. $k \to \infty$ or $\delta \to 0$). When we write statements such as

$$f(k) = C\,k^{-\beta}\big(1 + o(1)\big) \quad (k \to \infty),$$

the dependence of the $o(1)$ term on other parameters (e.g. $N, \delta$) will be clear from context or specified explicitly.

We reserve the symbol $\sim$ for asymptotic equality in the sense above and *do not* use the informal symbol "$\approx$" in our main statements. Informal approximations are always tied either to an explicit limit (e.g. $k \to \infty$, $\delta \to 0$) or to a pointer to a precise asymptotic statement in the appendices.

**Local log–log slopes.** For a fixed $k$-window $[k_1, k_2]$ and fixed $N$, we define the empirical *effective inference exponent* as the local log–log slope

$$\beta_{\text{eff}}(N; [k_1, k_2]) := -\frac{\log \mathcal{L}_{\text{inf}}(k_2; N) - \log \mathcal{L}_{\text{inf}}(k_1; N)}{\log k_2 - \log k_1}. \tag{23}$$

In experiments we take $[k_1, k_2]$ to be a central window where the log–log curve for $\mathcal{L}_{\text{inf}}(\cdot; N)$ is approximately linear; the precise window does not affect the qualitative behavior of $\beta_{\text{eff}}(N)$.

## C  HIGH DIMENSIONAL RIDGE REGRESSION WITH NOISY LABELS

In this appendix, we re-derive and analyze the generalization error for high-dimensional Ridge linear regression with additive label noise used in the main text. While these results are well-established in the literature (see, e.g., Maloney et al. (2022); Hastie et al. (2020) and references therein), we present a concise derivation to highlight the connection to our Latent Instance Difficulty (LID) model and to set the stage for understanding the training scaling laws discussed in Section 4.1.

### C.1  MODEL SETUP

We consider a linear model where the learner aims to estimate a true underlying parameter vector $\boldsymbol{\theta}^* \in \mathbb{R}^d$ from $N$ training samples $(\mathbf{x}_i, y_i)$. The input features $\mathbf{x}_i \in \mathbb{R}^d$ are drawn i.i.d. from a distribution with zero mean and covariance matrix $\boldsymbol{\Sigma} = \mathbb{E}[\mathbf{x}\mathbf{x}^\top]$. The observed labels $y_i$ are generated according to

$$y_i = \mathbf{x}_i^\top \boldsymbol{\theta}^* + \eta_i, \tag{24}$$

where $\eta_i$ is the label noise for sample $i$. In the context of our LID model (Definition 3.1), $\eta_i$ is a realization from $\mathcal{N}(0, \sigma_\eta^2/\tau_{\mathbf{x}_i})$. For the present derivation, we assume $\eta_i$ are i.i.d. with $\mathbb{E}[\eta_i] = 0$ and $\mathbb{E}[\eta_i^2] = \sigma_{\text{noise}}^2 < \infty$. If we were directly applying LID, $\sigma_{\text{noise}}^2$ would be replaced by $\mathbb{E}[\sigma_\eta^2/\tau_{\mathbf{x}}] = \sigma_\eta^2 \mathbb{E}[1/\tau_{\mathbf{x}}]$.

The learner estimates $\boldsymbol{\theta}^*$ using Ridge regression by minimizing the loss

$$L(\hat{\boldsymbol{\theta}}) = \frac{1}{2N} \sum_{i=1}^N (y_i - \mathbf{x}_i^\top \hat{\boldsymbol{\theta}})^2 + \frac{\lambda}{2}\|\hat{\boldsymbol{\theta}}\|_2^2, \tag{25}$$

where $\lambda \ge 0$ is the regularization parameter. Let $X \in \mathbb{R}^{N \times d}$ be the matrix of training features (rows $\mathbf{x}_i^\top$) and $\mathbf{y} \in \mathbb{R}^N$ the vector of observed labels. The solution is

$$\hat{\boldsymbol{\theta}}_\lambda = \Big(\frac{1}{N} X^\top X + \lambda \mathbf{I}_d\Big)^{-1} \frac{1}{N} X^\top \mathbf{y}. \tag{26}$$

## C.2 GENERALIZATION ERROR

The generalization error (or test loss) measures the expected squared prediction error on unseen test data $\mathbf{x}_{\text{test}}$ with true mean target $\mathbf{x}_{\text{test}}^\top \boldsymbol{\theta}^*$:

$$\mathcal{L}_{\text{gen}}(N, \lambda) = \mathbb{E}_{\mathcal{D}_N, \mathbf{x}_{\text{test}}} \left[ \left( \mathbf{x}_{\text{test}}^\top \hat{\boldsymbol{\theta}}_\lambda - \mathbf{x}_{\text{test}}^\top \boldsymbol{\theta}^* \right)^2 \right]. \tag{27}$$

Let $\Delta\boldsymbol{\theta} := \hat{\boldsymbol{\theta}}_\lambda - \boldsymbol{\theta}^*$. Then $\mathcal{L}_{\text{gen}} = \mathbb{E}[\Delta\boldsymbol{\theta}^\top \boldsymbol{\Sigma} \Delta\boldsymbol{\theta}]$, where the expectation is over the training data $\mathcal{D}_N$. Substituting $\mathbf{y} = X\boldsymbol{\theta}^* + \vec{\eta}$ (where $\vec{\eta}$ is the vector of noise realizations $\eta_i$) into equation 26, we obtain

$$\hat{\boldsymbol{\theta}}_\lambda = \left( \frac{1}{N} X^\top X + \lambda \mathbf{I}_d \right)^{-1} \frac{1}{N} X^\top (X\boldsymbol{\theta}^* + \vec{\eta})$$

$$= \left( \frac{1}{N} X^\top X + \lambda \mathbf{I}_d \right)^{-1} \left( \frac{1}{N} X^\top X\boldsymbol{\theta}^* + \frac{1}{N} X^\top \vec{\eta} \right).$$

Thus the error vector is

$$\Delta\boldsymbol{\theta} = \hat{\boldsymbol{\theta}}_\lambda - \boldsymbol{\theta}^*$$

$$= \left( \frac{1}{N} X^\top X + \lambda \mathbf{I}_d \right)^{-1} \left( \frac{1}{N} X^\top X\boldsymbol{\theta}^* + \frac{1}{N} X^\top \vec{\eta} \right) - \boldsymbol{\theta}^*$$

$$= \left( \frac{1}{N} X^\top X + \lambda \mathbf{I}_d \right)^{-1} \left[ \frac{1}{N} X^\top \vec{\eta} - \lambda \boldsymbol{\theta}^* \right]. \tag{28}$$

Assuming $\boldsymbol{\theta}^*$ is fixed (not random) and $\mathbb{E}[\vec{\eta}] = \mathbf{0}$, $\mathbb{E}[\vec{\eta}\vec{\eta}^\top] = \sigma_{\text{noise}}^2 \mathbf{I}_N$,

$$\mathbb{E}[\Delta\boldsymbol{\theta}] = -\left( \frac{1}{N} X^\top X + \lambda \mathbf{I}_d \right)^{-1} \lambda \boldsymbol{\theta}^*$$

and

$$\text{Cov}(\Delta\boldsymbol{\theta}) = \left( \frac{1}{N} X^\top X + \lambda \mathbf{I}_d \right)^{-1} \mathbb{E}\left[ \left( \frac{1}{N} X^\top \vec{\eta} \right) \left( \frac{1}{N} X^\top \vec{\eta} \right)^\top \right] \left( \frac{1}{N} X^\top X + \lambda \mathbf{I}_d \right)^{-1}$$

$$= \left( \frac{1}{N} X^\top X + \lambda \mathbf{I}_d \right)^{-1} \frac{\sigma_{\text{noise}}^2}{N^2} X^\top X \left( \frac{1}{N} X^\top X + \lambda \mathbf{I}_d \right)^{-1}. \tag{29}$$

Hence

$$\mathcal{L}_{\text{gen}} = \mathbb{E}[\Delta\boldsymbol{\theta}]^\top \boldsymbol{\Sigma} \mathbb{E}[\Delta\boldsymbol{\theta}] + \text{Tr}\left( \boldsymbol{\Sigma} \, \text{Cov}(\Delta\boldsymbol{\theta}) \right)$$

$$= \lambda^2 \, \boldsymbol{\theta}^{*\top} \left( \hat{\boldsymbol{\Sigma}} + \lambda \mathbf{I}_d \right)^{-1} \boldsymbol{\Sigma} \left( \hat{\boldsymbol{\Sigma}} + \lambda \mathbf{I}_d \right)^{-1} \boldsymbol{\theta}^*$$

$$+ \sigma_{\text{noise}}^2 \, \text{Tr}\left( \boldsymbol{\Sigma} \left( \hat{\boldsymbol{\Sigma}} + \lambda \mathbf{I}_d \right)^{-1} \hat{\boldsymbol{\Sigma}} \left( \hat{\boldsymbol{\Sigma}} + \lambda \mathbf{I}_d \right)^{-1} \right), \tag{30}$$

where $\hat{\boldsymbol{\Sigma}} = \frac{1}{N} X^\top X$ is the empirical covariance.

### C.2.1 UNDERPARAMETERIZED REGIME ($N \gg d$)

In the underparameterized limit $N \gg d$ with $\lambda \to 0$, the bias term vanishes and we obtain

$$\mathcal{L}_{\text{gen}} = \sigma_{\text{noise}}^2 \, \text{Tr}\left( \boldsymbol{\Sigma} \, \hat{\boldsymbol{\Sigma}}^{-1} \right). \tag{31}$$

In the setting discussed in the main text, the population covariance can be written as $\boldsymbol{\Sigma} = \boldsymbol{\Lambda}$, where $\boldsymbol{\Lambda}$ is diagonal with a decaying power-law spectrum as in equation 1. As stated in Silverstein and Bai (1995); Couillet and Liao (2022), the empirical covariance matrix can be expressed as $\hat{\boldsymbol{\Sigma}} = \boldsymbol{\Lambda}\mathbf{W}$ where $\mathbf{W}$ is a Wishart matrix with aspect ratio $\kappa = d/N$. Then

$$\mathcal{L}_{\text{gen}} = \sigma_{\text{noise}}^2 \, \text{Tr}\left( \mathbf{W}^{-1} \right). \tag{32}$$

The eigenvalues of $\mathbf{W}^{-1}$ follow the inverse Marchenko–Pastur law (Couillet and Liao, 2022), which implies

$$\mathbb{E}\left[ \text{Tr}(\mathbf{W}^{-1}) \right] = \frac{d}{N - d - 1} \qquad \text{for } N > d + 1. \tag{33}$$

Thus, in the high-sample limit $N/d \to \infty$,

$$\mathcal{L}_{\text{gen}}(N) \sim \sigma_{\text{noise}}^2 \frac{d}{N} \qquad (N \to \infty,\ N \gg d), \tag{34}$$

which recovers the $d/N$ tail used in equation 10 up to a constant factor.

Near the interpolation threshold $N \approx d$, one obtains the familiar peak in $\mathcal{L}_{\text{gen}}$ that forms one side of "double descent" (Belkin et al., 2019).

### C.2.2 OVERPARAMETERIZED REGIME ($N \ll d$)

In the overparameterized regime, the interpolation solution is obtained by taking a vanishing ridge parameter $\lambda \to 0$ with the scaling $\lambda = \tilde{\lambda}/N$ (Bahri et al., 2021). Under this scaling, the test loss decomposes as

$$\mathcal{L}_{\text{gen}} = \frac{\tilde{\lambda}^2}{N^2} \boldsymbol{\theta}^{*\top} \left( \hat{\boldsymbol{\Sigma}} + \tfrac{\tilde{\lambda}}{N} \mathbf{I}_d \right)^{-1} \boldsymbol{\Sigma} \left( \hat{\boldsymbol{\Sigma}} + \tfrac{\tilde{\lambda}}{N} \mathbf{I}_d \right)^{-1} \boldsymbol{\theta}^*$$
$$+ \sigma_{\text{noise}}^2 \operatorname{Tr}\left( \boldsymbol{\Sigma} \left( \hat{\boldsymbol{\Sigma}} + \tfrac{\tilde{\lambda}}{N} \mathbf{I}_d \right)^{-1} \hat{\boldsymbol{\Sigma}} \left( \hat{\boldsymbol{\Sigma}} + \tfrac{\tilde{\lambda}}{N} \mathbf{I}_d \right)^{-1} \right). \tag{35}$$

The first term (the bias) can be analyzed via replica or random matrix methods, leading to self-consistent equations of the form

$$\text{Bias}^2 = \frac{\tilde{\lambda}^2}{N^2} \sum_{i=1}^{d} \frac{\sigma_i^2}{\left( \sigma_i^2 + \tfrac{\tilde{\lambda}}{N} \right)^2}, \qquad \tilde{\lambda} = \sum_{i=1}^{d} \frac{\tfrac{\tilde{\lambda}}{N} \sigma_i^2}{\sigma_i^2 + \tfrac{\tilde{\lambda}}{N}}, \tag{36}$$

where $\{\sigma_i^2\}$ are the eigenvalues of $\boldsymbol{\Sigma}$; see Bordelon et al. (2020) for a detailed derivation. For power-law spectra $\sigma_i^2 \propto i^{-(1+\alpha)}$, solving equation 36 yields the scaling

$$\mathcal{L}_{\text{gen}}(N) \propto N^{-\alpha} \qquad (N \ll d),$$

up to multiplicative constants depending on the spectrum and on teacher alignment (Bartlett et al., 2020; Hastie et al., 2020). The noise term in the overparameterized regime behaves similarly to the underparameterized case with the roles of $d$ and $N$ exchanged, and contributes primarily near the interpolation threshold, completing the double-descent picture (Maloney et al., 2022).

In the context of the LID model, $\sigma_{\text{noise}}^2$ is replaced by $\sigma_\eta^2 \mathbb{E}[1/\tau_\mathbf{x}]$. Thus, Assumption 3.3 ($\beta > 2$) is crucial for these scalings to hold with finite prefactors; if $\beta \le 2$, $\mathbb{E}[1/\tau_\mathbf{x}]$ diverges and the standard Ridge regression analysis must be modified.

## D  INFERENCE SCALING AT FIXED $N$

We provide the detailed steps for the asymptotic evaluation of the pass@$k$ failure probability integral given in Eq. (11) for large $k$

$$\mathcal{L}_{\text{inf}}(k) = \mathbb{E}_{\tau_\mathbf{x} \sim \text{Gamma}(\beta/2,1)} \left[ [p(\mathbf{x}, \tau_\mathbf{x})]^k \right] \approx \mathbb{E}_{\tau_\mathbf{x} \sim \text{Gamma}(\beta/2,1)} [e^{-kc_\delta \sqrt{\tau_\mathbf{x}}}]. \tag{37}$$

Here, $c_\delta = 2\delta/(\sqrt{2\pi}\sigma_\eta)$ and $p(\mathbf{x}, \tau_\mathbf{x}) \approx 1 - c_\delta \sqrt{\tau_\mathbf{x}}$ for small $\tau_\mathbf{x}$. The PDF of $\tau_\mathbf{x} \sim \text{Gamma}(\beta/2, 1)$ is $f(\tau) = \frac{1}{\Gamma(\beta/2)} \tau^{\beta/2-1} e^{-\tau}$. The expectation integral is

$$\mathcal{L}_{\text{inf}}(k) = \int_0^\infty e^{-kc_\delta \sqrt{\tau}} \frac{1}{\Gamma(\beta/2)} \tau^{\beta/2-1} e^{-\tau} d\tau. \tag{38}$$

For large $k$, the factor $e^{-kc_\delta \sqrt{\tau}}$ decays extremely rapidly for any $\tau > 0$, forcing the dominant contribution to the integral to come from the region near $\tau = 0$. In this region, the term $e^{-\tau}$ in the Gamma PDF is approximately 1. Thus, we approximate the integral as

$$\mathcal{L}_{\text{inf}}(k) \approx \frac{1}{\Gamma(\beta/2)} \int_0^\infty e^{-kc_\delta \sqrt{\tau}} \tau^{\beta/2-1} d\tau. \tag{39}$$

We perform a change of variable: Let $u = kc_\delta\sqrt{\tau}$. Then $\sqrt{\tau} = u/(kc_\delta)$, which implies $\tau = (u/(kc_\delta))^2 = u^2/(kc_\delta)^2$. The differential is $d\tau = \frac{2u}{(kc_\delta)^2}du$. Substituting these into the integral equation 39

$$\int_0^\infty e^{-u}\left(\frac{u^2}{(kc_\delta)^2}\right)^{\beta/2-1}\frac{2u}{(kc_\delta)^2}du = \int_0^\infty e^{-u}\frac{u^{\beta-2}}{(kc_\delta)^{\beta-2}}\frac{2u}{(kc_\delta)^2}du$$

$$= \frac{2}{(kc_\delta)^{\beta-2}(kc_\delta)^2}\int_0^\infty e^{-u}u^{\beta-1}du$$

$$= \frac{2}{(kc_\delta)^\beta}\int_0^\infty u^{\beta-1}e^{-u}du.$$

The remaining integral is the definition of the Gamma function, $\Gamma(\beta)$, which converges for $\beta > 0$.

$$\int_0^\infty u^{\beta-1}e^{-u}du = \Gamma(\beta). \tag{40}$$

Substituting this back into the expression for $\mathcal{L}_{\text{inf}}(k)$

$$\mathcal{L}_{\text{inf}}(k) \approx \frac{1}{\Gamma(\beta/2)}\frac{2\Gamma(\beta)}{(kc_\delta)^\beta} = \left(\frac{2\Gamma(\beta)}{\Gamma(\beta/2)(c_\delta)^\beta}\right)k^{-\beta}. \tag{41}$$

This confirms the asymptotic scaling $\mathcal{L}_{\text{inf}}(k) \sim k^{-\beta}$ for large $k$.

### D.1 ASSUMPTIONS FOR THE TAUBERIAN STEP

We use standard Laplace–Stieltjes Tauberian arguments under the following mild conditions:

1. **Regularly varying difficulty near zero.** The latent precision satisfies $\Pr(\tau_x \leq t) = t^{\beta/2}L(t)$ as $t \downarrow 0$, with $L$ slowly varying and bounded on $(0, t_0]$.

2. **Uniform small-window expansion.** For some $c_\delta = \sqrt{2/\pi}\,\delta/\sigma_\eta$ and any compact $[\tau_{\text{lo}}, \tau_{\text{hi}}] \subset (0, \infty)$,

$$s(B, \tau) := \Pr\left(|B - \eta| \leq \delta \mid \tau\right) = c_\delta\sqrt{\tau}\exp\left(-\frac{B^2\tau}{2\sigma_\eta^2}\right)(1 + o(1))$$

   uniformly in $\tau \in [\tau_{\text{lo}}, \tau_{\text{hi}}]$ as $\delta \downarrow 0$, with $c_\delta = \sqrt{2/\pi}\,\delta/\sigma_\eta$ and $\eta \sim \mathcal{N}(0, \sigma_\eta^2/\tau)$.

3. **Model error.** $B_N(x) = x^\top(\hat{\theta}_\lambda - \theta^*)$ is centered sub-Gaussian with $\text{Var}[B_N] \asymp \mathcal{L}_{\text{gen}}(N)$ and is independent of $\tau_x$ (or weakly dependent so that conditioning on $\tau_x$ preserves sub-Gaussian tails).

4. **Independent trials and perfect verification.** Conditional on $(x, \tau_x)$ and the training set, the $k$ comparisons are i.i.d., and success is declared if any trial lies within the tolerance.

Under (1)–(4), Karamata-type Tauberian theorems yield the mixture law in equation 14 and the $k^{-\beta}$ tail in equation 12, with constants as stated.

### D.2 EQUIVALENCE TO MODEL-DRAW PASS@$k$.

Consider the alternative protocol that draws $k$ i.i.d. model proposals $\tilde{y}_j = \hat{y} + \xi_j$ with proposal noise $\xi$ having a bounded, smooth density $f_\xi$ independent of $\tau_x$, and verifies against a fixed target $y^\star$. Writing $B_N(x) = \hat{y} - m(x)$, the per-trial success probability is

$$\int_{|u|\leq\delta} f_\xi(B_N(x) - u)\,du = 2\delta\,f_\xi(B_N(x))(1 + o(1)) \quad (\delta \downarrow 0). \tag{42}$$

In our target-draw protocol the corresponding quantity is $2\delta\,f_\eta(B_N(x); \tau_x)$ with $f_\eta(\cdot; \tau_x) = \mathcal{N}(0, \sigma_\eta^2/\tau_x)$. Thus both protocols reduce to $(1-p)^k$ with $p \propto 2\delta$ times a *local density at* $B_N(x)$; since the only heavy tail in LID comes from $f_\eta(\cdot; \tau_x) \propto \sqrt{\tau_x}$ as $\tau_x \downarrow 0$, the small-success behavior, and hence the $k^{-\beta}$ tail, is unchanged by swapping model vs. target sampling (up to prefactors). This equivalence holds provided $f_\xi$ is bounded and does not itself introduce a $\tau_x$-dependent tail, and trials are conditionally independent.

# E  TRAINING-DEPENDENT INFERENCE SCALING IN THE LID LINEAR MODEL

We provide a more detailed derivation of the two-tail mixture law (Theorem 4.3) and the training-dependent effective exponent (Corollary 4.4).

## E.1  SETUP

Recall the LID setting:

- Features $\mathbf{x} \in \mathbb{R}^d$ are drawn from $\mathcal{N}(\mathbf{0}, \boldsymbol{\Sigma})$ with eigenvalues $\sigma_j^2 \propto j^{-(1+\alpha)}$.

- The teacher is linear: $m(\mathbf{x}) = \mathbf{x}^\top \boldsymbol{\theta}^*$.

- Each instance has latent precision $\tau_{\mathbf{x}} \sim \Gamma(\beta/2, 1)$, independent of $\mathbf{x}$.

- The stochastic target is $Y_{\mathbf{x}}^* \sim \mathcal{N}\big(m(\mathbf{x}), \sigma_\eta^2/\tau_{\mathbf{x}}\big)$.

- Training observes i.i.d. pairs $(\mathbf{x}_i, y_i)$ with $y_i \sim Y_{\mathbf{x}_i}^*$ and fits ridge/OLS to obtain $\hat{\boldsymbol{\theta}}_\lambda(N)$ from $N$ samples.

At test time we evaluate

- the training loss $\mathcal{L}_{\text{gen}}(N) = \mathbb{E}_{\mathbf{x}}[(\mathbf{x}^\top \hat{\boldsymbol{\theta}}_\lambda - \mathbf{x}^\top \boldsymbol{\theta}^*)^2]$,

- the inference loss $\mathcal{L}_{\text{inf}}(k; N)$ under a perfect verifier with tolerance $\delta > 0$,

where for each test $\mathbf{x}$ we draw $k$ i.i.d. $y_j \sim Y_{\mathbf{x}}^*$ and declare success if $\min_{j \leq k} |\mathbf{x}^\top \hat{\boldsymbol{\theta}}_\lambda - y_j| \leq \delta$.

## E.2  ASSUMPTIONS FOR THE TAUBERIAN ANALYSIS

We use standard Laplace–Stieltjes Tauberian arguments under the following conditions:

**Assumption E.1** (Difficulty tail and small-window success expansion).

1. **Near-zero tail of difficulty.** The latent precision has a regularly varying CDF

$$\Pr(\tau_{\mathbf{x}} \leq t) = t^{\beta/2} L(t) \quad \text{as } t \downarrow 0,$$

with $L$ slowly varying and $\beta > 0$. (For $\Gamma(\beta/2, 1)$ one has $L(t) \to 1/\Gamma(\beta/2 + 1)$.)

2. **Small-window success form.** For tolerance $\delta > 0$ and any compact $[\tau_{\text{lo}}, \tau_{\text{hi}}] \subset (0, \infty)$, there exists

$$c_\delta = \sqrt{\frac{2}{\pi}} \frac{\delta}{\sigma_\eta}$$

such that

$$s(B, \tau_{\mathbf{x}}) := \mathbb{P}(|B - \eta| \leq \delta \mid \tau_{\mathbf{x}}) = c_\delta \sqrt{\tau_{\mathbf{x}}} \, \exp\!\Big(-\frac{B^2 \tau_{\mathbf{x}}}{2\sigma_\eta^2}\Big)\big(1 + o(1)\big) \qquad (43)$$

as $\delta \downarrow 0$, uniformly in $\tau_{\mathbf{x}} \in [\tau_{\text{lo}}, \tau_{\text{hi}}]$, where $\eta \sim \mathcal{N}(0, \sigma_\eta^2/\tau_{\mathbf{x}})$.

**Assumption E.2** (Model error distribution). The prediction bias $B_N(\mathbf{x}) = \mathbf{x}^\top(\hat{\boldsymbol{\theta}}_\lambda - \boldsymbol{\theta}^*)$ satisfies:

- $\mathbb{E}[B_N(\mathbf{x})] = 0$,

- $B_N(\mathbf{x})$ is sub-Gaussian with $\text{Var}[B_N(\mathbf{x})] \asymp \mathcal{L}_{\text{gen}}(N)$,

- $B_N(\mathbf{x})$ is independent of $\tau_{\mathbf{x}}$ (or weakly dependent so that conditioning on $\tau_{\mathbf{x}}$ preserves sub-Gaussian tails).

Assumption E.1 is a mild regular-variation and smoothness condition. Assumption E.2 holds for the linear Ridge setting with Gaussian features and power-law covariance (App. C), where $B_N(\mathbf{x})$ is approximately Gaussian with variance scaling as $\mathcal{L}_{\text{gen}}(N)$.

### E.3 Two-tail law for single-try success probabilities

Define the single-try success probability

$$S_N(\mathbf{x}, \tau_\mathbf{x}) := 1 - p(\mathbf{x}, \tau_\mathbf{x}) = \mathbb{P}(|B_N(\mathbf{x}) - \eta| \le \delta \mid \tau_\mathbf{x}),$$

and let $S_N$ denote the random variable obtained by sampling $(\mathbf{x}, \tau_\mathbf{x})$ from the test distribution. We first characterize the small-$S_N$ tail of its distribution.

**Proposition E.3** (Two-tail law for $S_N$). *Under Assumptions E.1–E.2, there exist constants $A > 0$, $B(N) > 0$ and a function $\gamma(N) > 0$ such that, as $s \downarrow 0$,*

$$\mathbb{P}(S_N \le s) = A s^\beta + B(N) s^{\gamma(N)}(1 + o(1)), \tag{44}$$

*with*

$$\gamma(N) = \Theta\left(\frac{1}{\mathrm{Var}[B_N(\mathbf{x})]}\right) = \Theta\left(\frac{1}{\mathcal{L}_{gen}(N)}\right), \tag{45}$$

*and*

$$A = \frac{1}{\Gamma(\beta/2 + 1)} c_\delta^{-\beta}, \qquad c_\delta = \sqrt{\frac{2}{\pi}} \frac{\delta}{\sigma_\eta}.$$

*Sketch.* For small $\tau_\mathbf{x}$, $B_N(\mathbf{x})$ is negligible in equation 43 and $S_N \sim c_\delta \sqrt{\tau_\mathbf{x}}$. Since $\Pr(\tau_\mathbf{x} \le t) \sim t^{\beta/2}/\Gamma(\beta/2 + 1)$ as $t \downarrow 0$, we have

$$\mathbb{P}(S_N \le s) \supset \mathbb{P}\left(\tau_\mathbf{x} \le (s/c_\delta)^2\right) \sim \frac{1}{\Gamma(\beta/2 + 1)} \left(\frac{s}{c_\delta}\right)^\beta,$$

giving the $A s^\beta$ term.

For moderate $\tau_\mathbf{x} = \Theta(1)$, small $S_N$ arises from large $|B_N(\mathbf{x})|$. Inverting equation 43 for fixed $\tau_\mathbf{x}$ shows that $S_N \le s$ corresponds to $|B_N(\mathbf{x})| \gtrsim \sqrt{(2\sigma_\eta^2/\tau_\mathbf{x}) \log(c_\delta \sqrt{\tau_\mathbf{x}}/s)}$. Since $B_N(\mathbf{x})$ is sub-Gaussian with variance $\mathrm{Var}[B_N(\mathbf{x})]$, standard tail bounds imply

$$\mathbb{P}(S_N \le s \mid \tau_\mathbf{x}) \approx s^{c(\tau_\mathbf{x})/\mathrm{Var}[B_N(\mathbf{x})]},$$

up to slowly varying factors, where $c(\tau_\mathbf{x})$ is bounded away from zero and infinity on $\tau_\mathbf{x} \in [\tau_{\mathrm{lo}}, \tau_{\mathrm{hi}}]$. Integrating over $\tau_\mathbf{x}$ in this moderate range yields the second term $B(N) s^{\gamma(N)}$ with $\gamma(N) \asymp 1/\mathrm{Var}[B_N(\mathbf{x})]$. Combining the contributions gives equation 44; see, e.g., Tauberian theorems for Laplace–Stieltjes transforms in regular-variation settings. □

### E.4 Mixture law for pass@$k$

Recall that, under our independence assumptions,

$$\mathcal{L}_{\mathrm{inf}}(k; N) = \mathbb{E}[(1 - S_N)^k].$$

This is a Laplace–Stieltjes transform evaluated at $k$. Applying standard Tauberian theory for such transforms with the tail form equation 44 yields:

**Corollary E.4** (Mixture law for $\mathcal{L}_{\mathrm{inf}}(k; N)$). *Under the assumptions of Proposition E.3, there exist constants $C_1 > 0$ and $C_2(N) > 0$ such that, as $k \to \infty$,*

$$\mathcal{L}_{inf}(k; N) = C_1 k^{-\beta} + C_2(N) k^{-\gamma(N)}(1 + o(1)), \tag{46}$$

*with $\gamma(N)$ as in equation 45 and*

$$C_1 = \frac{2 \Gamma(\beta)}{\Gamma(\beta/2)} c_\delta^{-\beta}, \qquad c_\delta = \sqrt{\frac{2}{\pi}} \frac{\delta}{\sigma_\eta}.$$

Corollary E.4 is the more detailed version of Theorem 4.3 in the main text.

### E.5 EFFECTIVE LOG–LOG SLOPE

Let $\beta_{\text{eff}}(N; [k_1, k_2])$ be the local log–log slope defined in equation 23. Substituting equation 46 and taking $k_1, k_2 \to \infty$ with $k_2/k_1$ fixed, one sees that whichever term in equation 46 dominates over $[k_1, k_2]$ determines the slope. This yields:

**Corollary E.5** (Training-dependent effective exponent, precise form). *Under the conditions of Corollary E.4, fix a window $[k_1, k_2]$ such that for each $N$ either the $k^{-\beta}$ term or the $k^{-\gamma(N)}$ term dominates over this window. Then, as $k_1 \to \infty$,*

$$\beta_{\text{eff}}(N; [k_1, k_2]) = \min\{\beta, \gamma(N)\} + o(1).$$

*In particular, if $\mathcal{L}_{gen}(N)$ decreases monotonically with $N$, then $\gamma(N)$ increases and $\beta_{\text{eff}}(N)$ is monotonically non-decreasing in $N$ and converges to $\beta$.*

Corollary E.5 is the formal counterpart of Corollary 4.4 in the main text and justifies the empirical saturation fit equation 17.

### E.6 SUMMARY

---

**Summary: finite-$N$ correction and training-dependent exponent**

**Single-trial success (small window).** For tolerance $\delta > 0$ and bias $B_N(\mathbf{x})$,

$$1 - p(\mathbf{x}, \tau_{\mathbf{x}}) = c_\delta \sqrt{\tau_{\mathbf{x}}} \exp\left(-\frac{B_N(\mathbf{x})^2 \tau_{\mathbf{x}}}{2\sigma_\eta^2}\right)\left(1 + o(1)\right), \quad c_\delta = \sqrt{\frac{2}{\pi}} \frac{\delta}{\sigma_\eta}.$$

**Two-tail mixture.** As $k \to \infty$,

$$\mathcal{L}_{\text{inf}}(k; N) = C_1 k^{-\beta} + C_2(N) k^{-\gamma(N)}\left(1 + o(1)\right), \qquad \gamma(N) = \Theta\left(\frac{1}{\mathcal{L}_{\text{gen}}(N)}\right),$$

with $C_1$ given explicitly above.
**Effective slope.** In any fixed $k$-window where a single tail dominates,

$$\beta_{\text{eff}}(N) = \min\{\beta, \gamma(N)\} + o(1) \quad \text{as } k_1 \to \infty,$$

and hence $\beta_{\text{eff}}(N)$ is an increasing function of $N$ that saturates at $\beta$ as the model becomes well trained.

---

# F CIFAR-10H LID: SETUP AND EVALUATION DETAILS

## F.1 EXPERIMENTAL SETUP

**Feature extraction.** We pass each image through a ResNet-18 pretrained on ImageNet and extract the penultimate-layer vector $\mathbf{z}_i \in \mathbb{R}^d$ with $d = 512$. These frozen features play the role of $\mathbf{x}$ in our LID analysis, and we fit only the linear head.

**Stochastic Targets from Human Labels.** The CIFAR-10H dataset provides, for each image, the distribution of labels assigned by multiple human annotators. For each image $\mathbf{x}_i$ (mapped to feature $\mathbf{z}_i$), we calculate the mean $m_i = \sum_{c=0}^{9} c \cdot P(\text{label} = c|\mathbf{x}_i)$ and variance $v_i = \sum_{c=0}^{9} (c - m_i)^2 \cdot P(\text{label} = c|\mathbf{x}_i)$ of these human label distributions. The key connection to our LID model is made by treating the target for our linear regressor as intrinsically stochastic. In order to control the signal to noise ratio, we introduce a scaling factor $s$ to account for the magnitude of the noise variance, which is controllable in the LID setting. The learning problem is then defined on rescaled quantities

$$\text{Rescaled features: } \mathbf{z}_i^{\text{scaled}} = \mathbf{z}_i/s, \qquad \text{Rescaled mean target: } m_i^{\text{scaled}} = m_i^{\text{orig}}/s. \qquad (47)$$

During training, for each rescaled feature vector $\mathbf{z}_i^{\text{scaled}}$, the target label $y_i$ is sampled from a Gaussian distribution centered at its rescaled mean, using the human variance

$$y_i \sim \mathcal{N}(\text{mean} = m_i^{\text{scaled}}, \text{variance} = v_i + \epsilon_v). \qquad (48)$$

where $m_i^{\text{scaled}}$ is the rescaled mean human label for the instance, $v_i$ is the variance of the human labels, and $\epsilon_v$ is a small constant (e.g., $10^{-6}$) to ensure non-zero variance. Here, $v_i$ plays a role analogous to $1/\tau_{\mathbf{x}_i}$ in our synthetic LID model, with high human label variance corresponding to a "difficult" instance (low effective precision).

**Model and Training.** We perform fine tuning by training a linear regression model $\hat{y} = \mathbf{z}^{\text{scaled},T}\hat{\boldsymbol{\theta}}$ to predict a scalar value from the rescaled features $\mathbf{z}^{\text{scaled}}$ extracted from the pretrained ResNet. Given the quadratic loss and linear model, we compute the optimal weights $\hat{\boldsymbol{\theta}}$ analytically using the Ridge regression solution with an effectively scaled regularization parameter $\lambda_{\text{eff}} = \lambda/s^2$

$$\hat{\boldsymbol{\theta}}_\lambda = (N^{-1}\mathbf{Z}_{\text{scaled}}^T\mathbf{Z}_{\text{scaled}} + \lambda_{\text{eff}}\mathbf{I})^{-1}N^{-1}\mathbf{Z}_{\text{scaled}}^T\mathbf{y}, \qquad (49)$$

where $\mathbf{Z}_{\text{scaled}}$ is the matrix of rescaled training features, and $\mathbf{y}$ is the vector of sampled noisy labels (from Eq. (48)).

**Evaluation metrics.** *Generalization loss $\mathcal{L}_{gen}$.* On a held-out set we compute the MSE between $\hat{y}_{\text{val}} = \mathbf{z}_{\text{val}}^{\text{scaled}\,T}\hat{\boldsymbol{\theta}}$ and the rescaled human mean $m_{\text{val}}^{\text{scaled}}$. (To compare across different $s$, multiply by $s^2$.)
*Inference failure $\mathcal{L}_{inf}(k; N)$.* For each validation point we draw $k$ i.i.d. realizations $y_j \sim \mathcal{N}(m_{\text{val}}^{\text{scaled}}, v_{\text{val}} + \epsilon_v)$ and declare success if $|\hat{y}_{\text{val}} - y_j| \leq \delta_{\text{eff}}$ for any $j$, with $\delta_{\text{eff}} = \delta/s$. We average the all-fail indicator over the set. This implements the perfect-verification assumption from Sec. 4.2.

# G  ADDITIONAL EXAMPLES FROM CIFAR-10H

Here, we provide some additional examples from the CIFAR-10H dataset, to illustrate the type of stochastic labels inherent in the data.

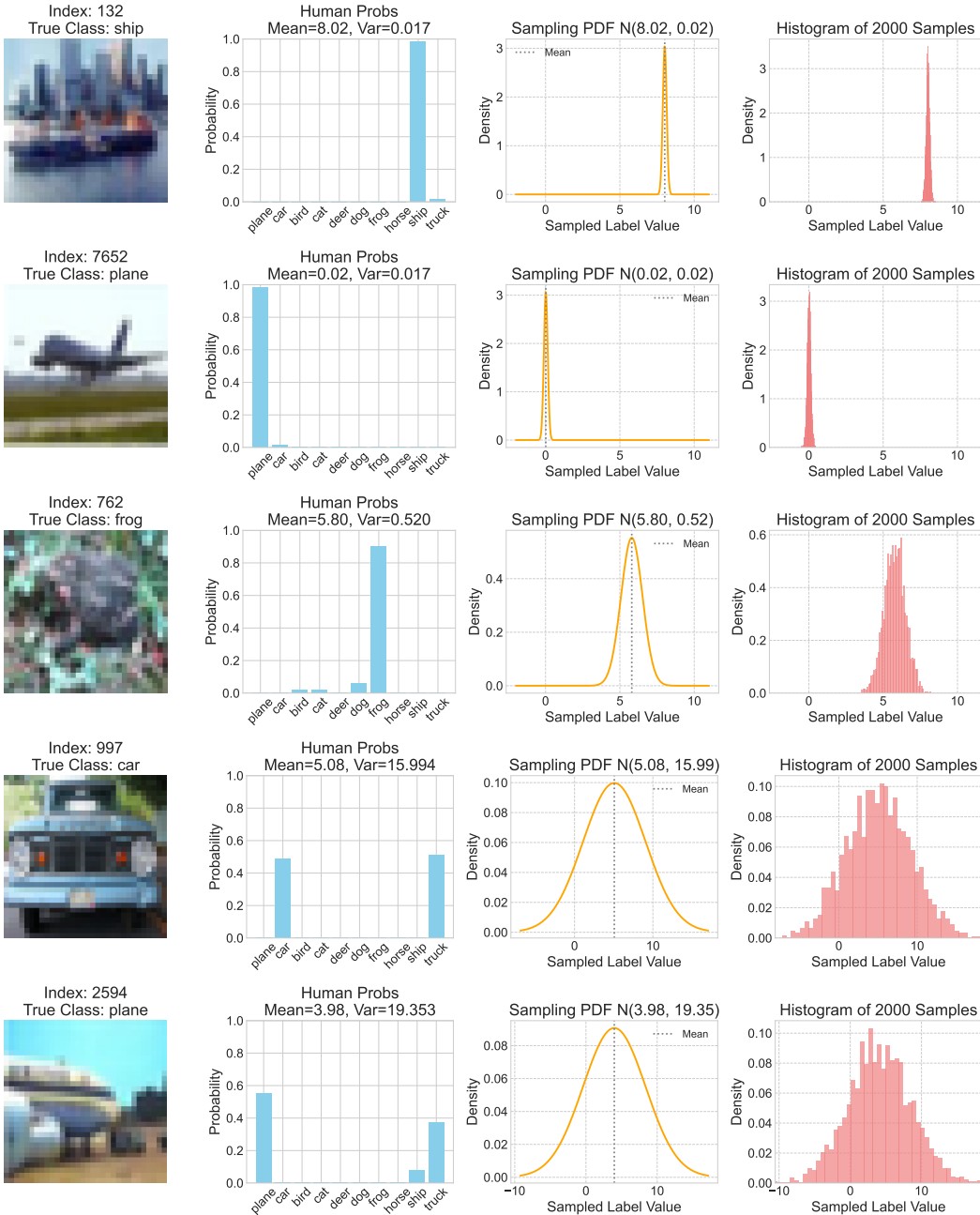

Figure 5: **Examples for low and high variance images from CIFAR-10H.** *Top to bottom:* low variance samples are easier to predict (more localized near the average prediction) while high variance samples are difficult.

## H GSM8K TEACHER–STUDENT DISTILLATION: SETUP AND EVALUATION DETAILS

**Data and models.** Teacher: `Flan-T5-XL`. Student: `Flan-T5-small` with LoRA (rank $r=8$, $\alpha=16$, dropout $0.05$) trained for one epoch, batch size 4, learning rate $5\times10^{-5}$. We sample one teacher solution per training question and fine-tune on subsets $N \in \{10, \dots, 6309\}$ (log-spaced). Evaluation uses the standard GSM8K test split.

**Strict greedy proxy for training loss.** For each test question we decode a single student output and parse a numeric only when it appears in an explicit "Final answer:" slot. Let $V_N$ be the set of test items with a valid parse. With tolerance $\delta = 10^{-3}$ we report

$$\mathcal{L}_{\text{gen}}^{(\text{greedy})}(N) = \frac{1}{|V_N|} \sum_{x \in V_N} \big(\hat{y}(x) - y^{\star}(x)\big)^2,$$

Winsorized by trimming the top $1\%$ of $|\hat{y} - y^{\star}|$ to reduce the influence of rare formatting/scale outliers. (Coverage $|V_N|/|\text{test}|$ is tracked but not plotted in the main text.)

**Inference metrics.** We compute $\mathcal{L}_{\text{inf}}(k; N) = 1 - \text{pass@}k$ using saved student samples and a perfect numeric verifier with the same tolerance $\delta$. The effective slope $\hat{\beta}_{\text{eff}}(N)$ is the local log–log slope of $\mathcal{L}_{\text{inf}}(\cdot; N)$ over a fixed central $k$ window (cf. §4.2).

**Interpretation and link to LID.** Distillation uses one noisy teacher sample per training input, so improvements can first appear as increased mass on good answers rather than a large shift in the point estimate. Consequently, $\mathcal{L}_{\text{inf}}(k; N)$ is more sensitive than a strict pass@1-style greedy metric. As $N$ grows, the finite-$N$ bias component shrinks, the $\mathcal{L}_{\text{inf}}(k; N)$ curves steepen, and $\hat{\beta}_{\text{eff}}(N)$ approaches the intrinsic difficulty index $\beta$ (Theorem 4.3).

**Limitations.** (i) The greedy metric is intentionally strict (explicit finals only), which undercounts improvements not emitted in canonical form; we therefore compute valid-only MSE. (ii) Using one teacher sample per training item does not directly target the teacher's mean; training-side MSE gains are modest but consistent with the theory's emphasis on inference-time behavior. (iii) Prefactors and saturation $N$ depend on LoRA capacity and teacher/student choice; the qualitative coupling of training and inference exponents is robust.

