# OpenReview forum: "Learning Shrinks the Hard Tail: Training‑Dependent Inference Scaling in a Solvable Linear Model"
_ICLR.cc/2026/Conference — ICLR 2026 Poster_

### Official Review · Reviewer_KXZD · 2025-11-01

**Soundness:** 3
**Presentation:** 3
**Contribution:** 3
**Rating:** 8
**Confidence:** 3

**Summary:**

This paper proposes the LID model, a simple analytical framework linking training-time and inference-time scaling laws. It shows that as training data scales, inference performance (pass@k) follows a steeper power-law w.r.t. k, with the effective exponent increasing and saturating at a limit—capturing how learning “shrinks the hard tail” of errors. The authors further derive a compute-allocation rule balancing training and inference effort and validate the theory through experiments on CIFAR-10H experiments.

**Strengths:**

- The paper introduces the LID model — a simple yet analytically tractable setup that connects training-time generalization scaling and inference-time scaling laws. This formalization is elegant and fills a clear theoretical gap between two rapidly developing empirical phenomena: training scaling laws and inference-time compute scaling.

- The derived theory provides clear insights into how the scale of training data affects inference scaling by effectively shrinking the hard tails of the error distribution. These are valuable conceptual contributions.

- The compute-allocation analysis (Section 4.3) offers interpretable results on when to invest in training versus inference compute, providing a practical takeaway.

- The paper also presents solid empirical results on CIFAR-10H that support the theoretical conclusions.

- The framework and analysis open up promising directions for future work, offering valuable foundations for studying the interaction between training-time scaling and test-time scaling.

**Weaknesses:**

The experiments are conducted on CIFAR-10, while inference-time scaling is a more relevant and currently critical topic for language models. It would be nice if some experiments could be conducted in that setting to better connect the work to practice.

**Questions:**

Please see the Weaknesses section.

---

> ### Author Response · Authors · 2025-11-13
>
> Dear reviewer KXZD,
>
>
> We thank the reviewer for recognizing the novelty of our contribution and acknowledging the rigor of our analysis, along with the solid experimental validation provided. We are glad that the reviewer regarded our contribution as filling a gap in the theory landscape and could open up new research directions regarding training/inference time scaling.
>
> Below, we address the reviewer's specific concern.
>
>  ### Weaknesses and Questions
> - **Experiments on LLMs** - We agree that additional experiments on LLMs would add value and extend our work to a more general setting. However, the goal of this work is to be a first proof of concept, showing that even in very simple settings, there can be a trade-off between training and sampling, which has been well noted by the reviewer and reflected in their evaluation.
> We will make an concerted effort to add LLM fine tuning results as soon as possible, but cannot guarantee that it can be done by the end of the rebuttal period, in which case we will add these results as an appendix to the final version.
>
>
> We thank the reviewer again for recommending to accept our paper, and would be happy to elaborate further or address additional questions as needed.

---

### Official Review · Reviewer_QTqr · 2025-11-01

**Soundness:** 3
**Presentation:** 1
**Contribution:** 3
**Rating:** 4
**Confidence:** 3

**Summary:**

This paper investigates the inference-time neural scaling laws specifically for the pass@$k$ metric. The analysis is grounded in a simple theoretical framework called the Latent Instance Difficulty (LID) model. Using this framework, the authors show that pass@$k$ follows a power law, where the exponent is shown to depend on both the generalization error of the trained model and the difficulty of the instance.

**Strengths:**

* This work provides a valuable theoretical analysis of inference-time neural scaling laws. A key contribution is establishing a novel connection, demonstrating how the scaling exponent is influenced by training-time dynamics.
* The paper introduces the Latent Instance Difficulty (LID) model, a simple yet valuable theoretical framework. This model is a notable contribution in its own right and shows potential for broader applicability beyond the immediate scope of this work.
* The theoretical contributions are well-supported by empirical validation on the Real-World CIFAR-10H dataset. This effectively bridges the gap between the proposed LID framework and real-world phenomena, demonstrating the practical relevance of the findings.

**Weaknesses:**

My main concern regards the presentation and rigor of the theoretical results. The current draft often discusses findings in a narrative, line-by-line fashion rather than consolidating them into complete, formal theorem statements. This fragmented presentation makes the paper's theoretical contributions difficult to follow and hard to assess.

In addition, the frequent usage of the approximation symbol ($\approx$) throughout the theoretical components, without a formal definition of the approximation level (e.g., in terms of asymptotic notation or explicit error bounds), makes the theoretical results feel incomplete and lack rigor.

Consolidating the key results (e.g., bounds, scaling exponents) into clearly stated theorems or propositions, along with rigorous definitions for all approximations used, would significantly improve the paper's clarity and trustworthiness.

I am open to increase my score if the presentation of this work can be improved during rebuttal period.

### Minors
* Please use LaTeX-style double quotes `` '' instead of " " (e.g., line 15, 20, 89).
* The metric 'pass@k' should be formatted in math mode pass@$k$ (e.g., line 16, 21).
* In line 134,137, 243, the authors use $Y_x^* \sim N(...)$ but $Y_x^* = N(...)$ would be more precise.
* It appears that every block equation has an equation number, even when not referenced. Please consider removing unnecessary equation numbers.

**Questions:**

* Could the authors elaborate on how these findings can be stated in a more rigorous manner? (See 'Weaknesses')
* Could the authors discuss on how the LID model might be applied in broader scenarios?

---

> ### Author Response · Authors · 2025-11-14
>
> Dear Reviewer QTqr,
>
> We sincerely thank you for your positive and very helpful review. We are extremely encouraged that you found our theoretical analysis valuable, the LID model a notable contribution, and the connection between training and inference novel.
>
> Below, we address your specific points as they were brought up.
>
> ### **Weaknesses**
> We accept your main concerns on the presentation and formal rigor of our theoretical results. Due to our academic background, we tend to prefer writing in a more narrative-driven style, but we understand how the informal use of approximations can make the results harder to assess than they should be. Following your guidance, we will implement all of your suggestions. Our working plan is as follows:
>
> **1. "Fragmented" Presentation of Theoretical Results:**
> We accept your assessment. Our findings should be consolidated into formal statements. To address this, we will restructure Section 4 and Appendix C to be much more rigorous:
> *   **Formal Propositions/Theorems:** We will consolidate our key findings into clearly stated, numbered Propositions or Corollaries. This will include:
>     *   A formal statement for the **asymptotic bias-free inference scaling** ($L_{inf} \sim k^{-\beta}$).
>     *   A formal proposition for the **two-tail mixture law** for $L_{inf}(k; N)$.
>     *   A formal corollary for the resulting **training-dependent effective exponent**, $\beta_{\mathrm{eff}}(N) \approx \min\{\beta, \gamma(N)\}$.
> This will make the theoretical contributions much clearer and easier to reference and evaluate.
>
> **2. On the Lack of Rigor with Approximations ($\approx$):**
> Thank you for bringing up this point. Our use of $\approx$ was indeed too informal. We will go through the manuscript and replace these with precise, formal statements.
> *   **Asymptotic Notation:** For asymptotic results (e.g., for large $k$), we will use standard asymptotic notation like $\sim$ (for leading-order behavior, e.g., $f(k) \sim C k^{-\beta}$ as $k \to \infty$) and $o(1)$ terms (e.g., $f(k) = C k^{-\beta} (1+o(1))$).
> *   **Explicit Conditions:** For approximations that are not strictly asymptotic (like the small-window expansion for success probability), we will either state the explicit error bounds or clearly define the regime in which the approximation holds (e.g., "in the limit $\delta \to 0$"). This is particularly relevant for the Tauberian argument assumptions, which we have already consolidated in Appendix C.1 but will ensure are clearly referenced.
>
> We have already begun implementing these structural changes and will upload a revised manuscript during the discussion period that reflects this more rigorous presentation.
>
> ### **Minors**
>
> Thank you for catching these formatting issues. We will correct all of them in the revised version:
> *   We will replace all instances of `" "` with LaTeX-style `` '' quotes.
> *   We will format `pass@k` in math mode as `pass@$k$` throughout the paper.
> *   We will use `\coloneqq` (or `:=`) for definitions as you suggested in lines 134, 137, and 243.
> *   We will use the `amsmath` environment `equation*` for unnumbered equations and remove unnecessary equation numbers to improve readability.
>
> We continue to answer your questions in the next response.

---

> > ### Author Response · Authors · 2025-11-14
> >
> > ### **Questions**
> >
> > **1. "Could the authors elaborate on how these findings can be stated in a more rigorous manner?"**
> > Certainly. As outlined above, the primary way we will do this is by consolidating our results into formal *Proposition* blocks. For example, the core result in Proposition 4.2 will be preceded by a formal statement of the two-tail mixture law, with clearly stated assumptions (drawing from Appendix C) regarding the regular variation of the difficulty distribution and the sub-Gaussian nature of the model's bias. All informal approximations will be replaced with precise asymptotic notation, making the conditions for each result explicit.
> >
> > **2. "Could the authors discuss on how the LID model might be applied in broader scenarios?"**
> > This is an important question. While our paper focuses on fine-tuning, the core idea that performance is a mixture of an intrinsic difficulty tail and a training-dependent error tail is likely a general principle.
> > *   **Beyond Fine-Tuning:** One could apply LID to analyze training from scratch, where the feature representations themselves are learned. In this case, not only does the linear head improve, but the features themselves become more effective, which might be modeled as the data spectrum $\alpha$ changing during training.
> > *   **Beyond Regression:** For classification, instance difficulty could be modeled by the "label noise" or ambiguity between two classes (e.g., a label distribution of [0.5, 0.5] for a cat-dog image has high intrinsic difficulty). The pass@k metric could then be applied to see if one of $k$ samples is assigned the correct class.
> > *   **Reinforcement Learning:** In RL, "instance difficulty" could correspond to the stochasticity of rewards in a given state. A state with highly variable rewards is intrinsically "harder" to learn a stable policy for.
> > *   **Reasoning on LLMs**: We expect to find the same kind of behavior beyond the LID in LLMs tested on reasoning tasks. We expect that the exponent determining the power law inference scaling of LLMs will improve in a qualitatively similar way when adding more tokens to pretraining, though this intuition is still mainly drawn from empirical results combined with this work.
> >
> > We will add a brief paragraph to the conclusion touching on these potential broader applications to highlight the generality of the LID conceptual framework.
> >
> > Thank you again for your highly constructive and applicable feedback. We are confident that the revised manuscript will meet your standards for presentation and rigor, improving the paper substantially and hopefully leading you to raise your score.

---

> > > ### Author Response · Authors · 2025-11-25
> > >
> > > Dear reviewer QTqr,
> > >
> > > We have uploaded a revised version where we have incorporated your suggestions on formalization, fixed all the typos we could find, included further discussions as well as an additional LLM distillation experiment on GSM8K.
> > >
> > > Thank you again for the useful comments and questions, and we hope that given the revised manuscript, you will consider raising your score. If you have any remaining concerns, we would be happy to engage further.

---

### Official Review · Reviewer_UZyf · 2025-11-01

**Soundness:** 3
**Presentation:** 3
**Contribution:** 3
**Rating:** 4
**Confidence:** 4

**Summary:**

The paper derives a training-dependent inference exponent and shows that the pass@k failure rate, decays as a power law, with an exponent that is small for poorly-trained models and grows with the number of training samples N, eventually saturating at an intrinsic limit determined by the tail of the task’s true difficulty distribution. This implies that when the model is undertrained, the marginal benefit of
acquiring more training data is high. Once the model is well-trained and the inference exponent has saturated, further gains are best sought by investing in more inference-time compute.

**Strengths:**

- Investigating the effect of the training data and its size on pass@k is an important and timely topic
- The conclusions are backed up with theoretical analysis
- The analysis yield concrete empirical guidance for training vs test-time compute

**Weaknesses:**

- The discussion of the analysis is deferred to the last page of the paper (and very briefly in page 7), this makes it hard to follow the derivatives and the conclusions. I suggest the authors discuss the conclusions after every derivation to clarify the messages throughout the paper. Besides, I strongly suggest the authors to revise the introduction and discuss the conclusions as it is done in the last page. This is very important for conveying the message to the readers, currently all the important messages are hidden in notations and equations. As someone who feels comfortable reading math, I had a hard time to understand the "importance" of the conclusions even after reading the intro for a couple of times.

- I liked the paper and its message and I have no problem with analysis in the simplified linear setting. However, to confirm the validity of and importance the conclusions in practice (after all, pass@k is mainly used for LLMs), why don't the authors add experiments with LLMs? This is probably the largest weakness of the paper.

**Questions:**

- Can you add experiments with LLMs to confirm the conclusions? I saw that you mentioned "For complex reasoning, a more powerful model might not only learn the mean better but also fundamentally simplify the problem, an effect our current model does not capture". Does it mean that you expect the conclusions to not hold in practice for LLMs? or you still believe the conclusions would hold? Confirming this in practice would add a big value to the paper.

ps: while experiments with LLMs are not cheap, fine-tuning smaller models is still doable in most academic labs.

---

> ### Author Response · Authors · 2025-11-13
>
> Dear reviewer UZyf,
>
> We thank you for the thoughtful and constructive review. We are very encouraged that you found the topic important and timely, the conclusions backed by theory, and the resulting guidance actionable. Your feedback on presentation and experimental validation is valuable, and we will revise the paper accordingly.
>
> Below, we address the specific points from your review.
>
> ### **Weaknesses**
>
> **1. On Presentation and Clarity:**
> You raise an excellent point about the presentation. We agree that the core intuitions and takeaways may indeed be too concentrated at the end of the paper, making it difficult to appreciate the importance of the results as they are being derived. This is a critical issue that we will fix.
>
> To address this, we will perform a significant revision of the paper's structure for clarity:
> *   **Revised Introduction:** As you suggest, we will revise the introduction to more clearly and directly state our main findings and their implications up front.
> *   **Takeaways Alongside Theory:** We will intersperse the theoretical sections (especially Section 4) with short, intuitive "Interpretation" or "Takeaway" paragraphs immediately following key derivations. For example, after deriving the $\beta_{\mathrm{eff}}(N)$ saturation, we will explicitly state in plain language what this means for a practitioner.
>
> We are confident these presentation changes alone will make the key messages much easier to parse and will significantly strengthen the paper.
>
> **2. On the Lack of LLM Experiments:**
> This is the most critical point, and we agree that connecting our theory to the LLM domain is important. This point is directly tied to your question, which we address in detail below.
>
> ### **Questions**
>
> **"Can you add experiments with LLMs to confirm the conclusions? ... Does [your limitation] mean that you expect the conclusions to not hold in practice for LLMs?"**
>
> Thank you for letting us expand on this subject. To be clear: **we absolutely believe the core qualitative conclusions of our model will hold for LLMs**, and your question has prompted us to design an experiment to demonstrate this.
>
> The limitation you quoted was intended to be a point of nuance and a clear future direction, not a refutation of our model's applicability. Let us clarify the distinction:
> 1.  **Our theory describes the learning process for a *fixed* model architecture.** It predicts that as you increase the size of the training data $N$, the model's effective inference exponent, $\beta_\mathrm{eff}(N)$, will improve and eventually saturate at a plateau, $\beta$. This $\beta$ represents the intrinsic difficulty of the task *as seen by that specific model*, restricted by its ability to solve tasks.
> 2.  The "simplification" effect you mention refers to what happens when you change to a **more powerful architecture**. A stronger model might indeed find the task fundamentally easier. In our framework, this would mean the powerful model has a *higher* intrinsic plateau, $\beta_\mathrm{powerful} > \beta_\mathrm{weak}$, which can depend, for instance, on the number of model parameters.
>
> However, the **saturation dynamic itself remains**. Our model provides the first theoretical framework to describe this fundamental dynamic, which we believe is a general principle of learning on tasks with heterogeneous difficulty.
>
> We believe that isolating and analyzing this single effect: how $\beta_\mathrm{eff}$ changes with $N$ for a fixed model, is a novel and valuable contribution in its own right, as it has never been explored theoretically.
>
> That being said, we fully agree that an LLM experiment would provide powerful, direct validation. We will make a strong effort to complete the following experiment during the rebuttal period:
> *   **Plan:** We will model the practical task of distilling a math reasoning task (from GSM8K) from an imperfect "teacher" LLM to a smaller "student" model. The variance in the teacher's answers will provide a natural measure of instance difficulty, allowing us to test if the student's $\beta_\mathrm{eff}(N)$ saturates as predicted.
>
> This experiment would directly validate our conclusions in a modern LLM context. We are working to include these results in a revised manuscript to be uploaded during the discussion period. Should we be unable to complete the full experiment in time, we will include the final results in the camera-ready version. We hope you'll agree that the paper's core theoretical contribution, once made clearer through our presentation revisions, is significant on its own.
>
> We believe these revisions: (i) greatly improving the paper's narrative clarity and (ii) potentially adding a LLM-based experiment, will fully address your concerns and substantially enhance the paper's contribution.
>
> Thank you again for your thoughtful comments and suggestions, we would be happy to answer any further questions you may have after we have uploaded a revised version.

---

> > ### Author Response · Authors · 2025-11-25
> >
> > Dear reviewer UZyf,
> >
> > We have uploaded a revised version where we have incorporated your suggestions, including an LLM distillation experiment on GSM8K.
> >
> > Thank you again for the useful comments and questions, and we hope that given the revised manuscript you will consider raising your score. If you have any remaining concerns, we would be happy to engage further.

---

### Author Response · Authors · 2025-12-02

Dear ACs/SACs,

We would first like to thank the reviewers for their useful comments and questions. Given the recent changes to the ICLR review process, we felt that a summary of the reviews and our rebuttals would benefit the AC/SAC when making their decision.
Below is a concise summary of (i) what the reviewers highlighted as strengths in the original submission, (ii) the main concerns they raised, and (iii) how our revision addresses those points and strengthens the paper.

_____
### **Strengths cited by all reviewers:**

* **Valuable theoretical contribution:** The paper offers a clear analysis of **inference‑time scaling** and establishes a **novel link** between training‑time generalization and the pass@$k$ exponent via the Latent Instance Difficulty (LID) model within the context of linear fine-tuning.
* **Conceptual clarity and portability:** The **LID framework** is simple, interpretable, and potentially applicable beyond the specific setting studied.
* **Empirical support:** Results on **CIFAR‑10H** substantiate the theory, demonstrating that the training–inference coupling and exponent saturation predicted by LID appear in real data when fine tuning a linear layer on features extracted from a pretrained model.

We would like to point out that both of the reviewers whose original score was $4$ indicated either directly or implicitly that they would lean towards raising their score if their concerns are met (e.g., QTqr: *``I am open to increase my score if the presentation of this work can be improved during rebuttal period.''*).
___

### **Main issues raised**

* **Formalized presentation:** Theory was sometimes presented in narrative manner rather than formal propositions/theorems.
* **Loose use of $\approx$:** Frequent use of approximation symbols without clear regimes, asymptotic statements, or error terms reduced perceived rigor in the opinion of one of the reviewers.
* **Scope & broader relevance:** A request to **clarify broader applicability** of LID (e.g., classification, RL, LLM reasoning, training from scratch), as well as inclusion of an LLM fine fine tuning experiment.
* **Formatting and style minor points:** Minor typesetting issues were pointed out (quotes instead of LaTeX “``…’’”, pass@$k$ not consistently in math mode, etc.).
____

### **How our revision addresses these concerns**

* **Formal structure**

   * We **reorganized the theory** into clearly stated **Propositions/Theorems/Corollaries** with explicit assumptions (regular variation, small‑window expansions, independence/sub‑Gaussianity) and a **Notation & asymptotics** subsection.
   * We replaced informal $\approx$ with **standard asymptotic notation** and **explicit regimes** (e.g., $f(k)=Ck^{-\beta}(1+o(1))$, use of $\sim$, $O$, $\Theta$); where appropriate we note **finite‑window fits** separately from asymptotic statements.
   * The **two‑tail mixture law** for pass@$k$ and the **training‑dependent effective exponent** $\beta_{\mathrm{eff}}(N)=\min{\beta,\gamma(N)}$ are now **formally stated** and cross‑referenced to assumptions and derivations.

* **Empirical clarity & additional evidence**

   * We preserved the **CIFAR‑10H** experiment but **streamlined** its main‑text description for space, moving full details to an **Appendix**; the figures continue to show the $1/N$ training tail, steepening pass@$k$ curves, and saturation of $\beta_{\mathrm{eff}}(N)$.
   * We added a **second, complementary GSM8K teacher–student distillation** testbed. This experiment probes a reasoning‑style setting and shows the same qualitative pattern: **training‑side improvement** accompanies **steeper pass@$k$** and a **saturating $\hat{\beta}_{\mathrm{eff}}(N)$**. We used a **strict numeric parsing** protocol and report **valid‑only losses** to avoid artifacts.

* **Formatting & style corrections**

   * We fixed all minor issues and inaccuracies following the reviewer's comments.

* **Broader applicability**

   * We added a short **Conclusions** paragraph outlining how LID extends conceptually to **training‑from‑scratch** (feature spectrum evolving with training), **classification** (instance ambiguity), **reinforcement learning** (reward stochasticity as difficulty), and **LLM reasoning** (pretraining scale affecting the effective tail), addressing the reviewer’s request.
______
### **Summary**

The revised manuscript is **sharper, more rigorous, and easier to verify**: key results are formal, approximations are precisely scoped, and experiments are more diverse. The **central claim**: that training shrinks the “hard tail,” raising $\beta_{\mathrm{eff}}(N)$ until it saturates at the intrinsic $\beta$, is now **better supported theoretically and empirically**.

We hope this summary clearly communicates that our changes directly address the reviewers’ concerns while preserving the original paper’s conceptual simplicity and contributions, leading to acceptance of our paper.

---

### Meta-Review · Area_Chair_PtaD · 2026-01-07

**Summary:**

This work proposes a theoretically solvable model for understanding the dependence of inference scaling on training. The concept of this model is concrete and its results are sufficiently suggestive, which no reviewers doubt.

 In the revised manuscript, the authors clarified the unrigorous notations pointed out by Reviewer QTqr and provided additional empirical confirmation, as suggested by Reviewer UZyf and Reviewer KXZD. I think that these modifications are sufficient to address the weaknesses identified by the reviewers identified by the reviewers and adequately resolve their concerns.

**Reviewer Concerns:**

Since Reviewer QTqr’s points were on presentation aspects, I think they have all been resolved by the revision.

Reviewer UZyf’s major concern regarding the weaknesses was the lack of experiments on LLMs. I think this was addressed, at least in a minimal form, by the additional experiment on GSM8K teacher–student distillation. This also resolves Reviewer KXZD’s concern about the experiments being limited to CIFAR-10.

**Reviewer Scores:**

Because of the resolved concerns as mentioned above, I expect that

*Reviewer QTqr would increase their score to 8. Note that they stated in their original review that “I am open to increasing my score if the presentation of this work can be improved during the rebuttal period.”

*Reviewer UZyf would increase their score to 6.

---

### Decision · Program_Chairs · 2026-01-26

Accept (Poster)